# Microbial metabolites tune amygdala neuronal hyperexcitability and anxiety-linked behaviors

Weonjin Yu [1,6], Yixin Xiao[1,6], Anusha Jayaraman [2,6], Yi-Chun Yen[1], Hae Ung Lee[2], Sven Pettersson [2,3,4,5✉] & H Shawn Je [1✉]

## Abstract

Changes in gut microbiota composition have been linked to anxiety behavior in rodents. However, the underlying neural circuitry linking microbiota and their metabolites to anxiety behavior remains unknown. Using male C57BL/6J germ-free (GF) mice, not exposed to live microbes, increased anxiety-related behavior was observed correlating with a significant increase in the immediate early c-Fos gene in the basolateral amygdala (BLA). This phenomenon coincided with increased intrinsic excitability and spontaneous synaptic activity of BLA pyramidal neurons associated with reduced small conductance calcium-activated potassium (SK) channel currents. Importantly, colonizing GF mice to live microbes or the microbial-derived metabolite indoles reverted SK channel activities in BLA pyramidal neurons and reduced the anxiety behavioral phenotype. These results are consistent with a molecular mechanism by which microbes and or microbial-derived indoles, regulate functional changes in the BLA neurons. Moreover, this microbe metabolite regulation of anxiety links these results to ancient evolutionarily conserved defense mechanisms associated with anxiety-related behaviors in mammals.

**Keywords** Gut Microbiota; Anxiety; Indoles; Neuronal Excitability; Amygdala
**Subject Categories** Microbiology, Virology & Host Pathogen Interaction; Neuroscience

## Introduction

Emotion is intimately linked to organic life (Letourneau, 1878) and anxiety is associated with physiological and behavioral changes that prepare an individual for threat. However, few studies have addressed the implications of this "being anxious" function without disease states. Anxiety is triggered as the body's adaptive response to stress by the hypothalamic-pituitary-adrenal (HPA) axis, one of the key components of the neuroendocrine system (Tsigos and Chrousos, 2002). The sustained activation of the HPA axis under continued stress results in elevated levels of stress hormones and the persistent behavioral changes seen in anxiety disorders (Faravelli et al, 2012). Several studies show that the thresholds and triggers for anxiety differ between mouse strains, as does the ability to cope with stress during malnutrition. For example, germ-free (GF) mice typically show increased HPA activity and elevated cortisol levels, although some GF mouse strains such as Balb/c do not show elevated cortisol levels despite the absence of living microbes. The underlying reasons for the differences between mouse strains remain unknown. In general, when GF mice are exposed to live microbes, the HPA axis signal is reduced, and cortisol levels are lowered (Sudo et al, 2004). Thus, the formation of a symbiotic relationship between the incoming microbes after birth and the newly found mice host appears to be an important factor in regulating anxiety.

The amygdala is a critical part of the neural circuitry underlying anxiety-related behavior (Millan, 2003). Anxiety or fear emotions trigger powerful survival signals by warning of potential danger, including nutrient deprivation (Kris-Etherton et al, 2021). The basolateral amygdala (BLA) communicates bidirectionally with brain regions including the prefrontal cortex, hippocampus, nucleus accumbens, and hindbrain regions that influence cognition, motivation, and stress responses. Hyperexcitability of the basolateral amygdala neurons has been associated with several stress-induced anxiety-related behaviors in rodents (Roozendaal et al, 2009).

Several studies in the past have reported a correlation between gut microbiota and behavior in mice (Diaz Heijtz et al, 2011; Gareau et al, 2011; Neufeld et al, 2011; Clarke et al, 2013; Crumeyrolle-Arias et al, 2014; Arentsen et al, 2015). Specifically, research carried out by us and others has demonstrated the important role of the gut microbiota in the regulation of anxiety-related behavior and brain chemistry in rodents (Park et al, 2013; Foster et al, 2016; Hoban et al, 2018; Martin et al, 2018; Flux and Lowry, 2020). The metabolites secreted by the microbes also regulate the blood-brain-barrier integrity (Braniste et al, 2014). Anxiety disorders are complex, heterogeneous, and the most common mental disorders (Bandelow and Michaelis, 2015; Chisholm et al, 2016; Remes et al, 2016). Clinically, gut microbes

[1]Program in Neuroscience and Behavioral Disorders, Duke-NUS Medical School, 8 College Road, Singapore 169857, Singapore. [2]ASEAN Microbiome Nutrition Centre, National Neuroscience Institute, 11 Jalan Tan Tock Seng, Singapore 308433, Singapore. [3]Karolinska Institutet, Department of Dental Medicine, Stockholm, Sweden. [4]School of Medical and Life Sciences, Sunway University, Sunway City 47500, Malaysia. [5]Department of Microbiology and Immunology, National University, Singapore, Singapore. [6]These authors contributed equally: Weonjin Yu, Yixin Xiao, Anusha Jayaraman. ✉E-mail: sven.pettersson@ki.se; shawn.je@duke-nus.edu.sg

have been implicated in the development of pathological anxiety, as evidenced by the high comorbidity rate between anxiety disorders and functional gastrointestinal (GI) disorders (Banerjee et al, 2017). Functional magnetic resonance imaging (fMRI) studies have shown amygdala hyperactivity in patients with anxiety disorders (Straube et al, 2006; Etkin and Wager, 2007). Moreover, the BLA glutamatergic excitatory neurons show increased intrinsic excitability during chronic early-life stress in rats (Rau et al, 2015). However, the causal relationship among gut microbiota, anxious behavior, and anxiety disorders is not clearly understood. To better understand this relationship, we combined data from behavioral analyses and electrophysiological recordings in C57BL/6J germ-free (GF) mice. C57BL/6J is the most widely used inbred strain to generate or back-cross genetically modified mice with fewer anxiety-related behaviors compared to other strains (Matsuo et al, 2010).

Here, we report that the male C57BL/6J GF mice exhibit anxiety-related behavior and increased intrinsic excitability in BLA pyramidal neurons. This increased neuronal excitability was due to a reduction in small conductance $K^+$ (SK) channel-dependent medium afterhyperpolarization (mAHP). Furthermore, these electrophysiological changes in the BLA principal neurons were reversed by introducing live microbes into the recipient GF mice or by treating them with indole, a microbiota metabolite, which restored normal behavior in the GF mice.

## Results

### Exhibition of anxiety-related behaviors in GF mice

To assess the locomotor activity phenotype and the associated anxiety-like behavior in either GF or SPF C57BL/6J male mice, we recorded the mice using an open-field test (Seibenhener and Wooten, 2015). We found that GF C57BL/6J mice exhibited reduced motor activity, as evidenced by the total distance traveled (GF: 49.11 ± 4.68; SPF: 71.05 ± 5.41 m; Student's $t$ test, $P = 0.0067$, Fig. 1A), compared to the SPF control. This finding contrasts with a previous report in GF mice derived from NMRI strains (GF NMRI) (Diaz Heijtz et al, 2011).

Using an elevated zero maze (EZM) test, we observed that the GF C57BL/6J mice spent less time in open quadrants (SPF: 42.05 ± 6.78 s; GF: 11.52 ± 3.69 s; Student's $t$ test, $P = 0.0008$) (Fig. 1B) and less frequently moved from one closed quadrant to the other (SPF: 3.89 ± 0.96; GF: 0.8 ± 0.36; Student's $t$ test, $P = 0.0061$) (Fig. 1C). Since less anxious mice tend to explore open areas in the EZM test, this finding indicates an elevated level of anxiety in the male C57/Bl6 GF mice. Taken together, these data suggest that GF C57BL/6J mice, unlike the NMRI mice (Diaz Heijtz et al, 2011), exhibit increased anxiety-related behavior compared to SPF controls.

### Identification of basolateral amygdala (BLA) neurons involved in anxiety-related behaviors of GF mice

We further investigated the neural circuit mechanisms contributing to the altered behavior in GF mice. In this experiment, we established two distinct control groups: the naive group, in which mice remained undisturbed in their home cages, and the EZM-exposed group, which included both GF and SPF C57BL/6J mice. Specifically, we sought to examine the brain regions activated by a single exposure to EZM. To accomplish this, we exposed both GF and SPF C57BL/6J mice, which were naive to EZM, to the maze for 30 min (Fig. 1D). We then performed immunofluorescence staining using specific antibodies against c-Fos, a widely recognized marker of neuronal activity due to its stereotypical expression in response to external stimuli (Kovacs, 2008; Chung, 2015).

Interestingly, we observed a notable increase in c-Fos-positive cells in several areas within the GF group. Specifically, within the amygdala, a critical component of the neural circuitry underlying anxiety-related behavior, we observed a significant increase in the number of c-Fos-positive cells within the basolateral amygdala (BLA), but not the central amygdala (CeA), of GF mice compared to SPF mice (SPF: 182.8 ± 24.62 n/mm²; GF: 343.8 ± 7.62 n/mm²; Student's $t$ test <0.0001) (Fig. 1E,I). In the naive group, we did not observe a statistically significant increase in c-Fos-positive cells in the BLA, CeA, or hippocampal CA1 areas within the GF group compared to SPF mice (SPF_BLA: 17.25 ± 1.10 n/mm²; GF_BLA: 22.5 ± 1.32 n/mm²; Mann–Whitney test $P = 0.06$) (Appendix Fig. S1A–C).

We also examined other areas previously implicated in anxiety-related behaviors, such as the bed nucleus of the stria terminalis (BNST), prefrontal cortex (PFC), and ventral hippocampal CA1 (vCA1) (Fig. 1F–I). While we did not observe a significant difference in the number of c-Fos-positive cells in the BNST or PFC, we did observe a higher number of c-Fos-positive cells in the prelimbic (PL) and infralimbic (IL) regions compared to other areas in both SPF and GF mice (Fig. 1I). Interestingly, we observed a slight, but not statistically significant increase in the number of c-Fos-positive cells in ventral CA1 cells (SPF: 35.5 ± 1.32; GF: 42 ± 2.67, Mann–Whitney test $P = 0.14$), consistent with the known reciprocal connection between the BLA and the ventral hippocampus (vHPC), which together form the control center for fear and anxiety-related behavior (Fig. 1H,I) (Felix-Ortiz et al, 2013). Taken together, these data suggest that BLA neurons may become active during anxiety-related behaviors in GF mice.

### Increased excitability of principal neurons in the BLA of GF mice

Next, we performed an electrophysiological recording of BLA principal neurons to assess the functional changes between GF and SPF mice (Fig. 2A). For this, we injected a series of increased step current pulses (−200 to 500 pA in 50 pA increments) into the neurons in the presence of CNQX, a glutamate receptor antagonist, and picrotoxin, a GABA receptor blocker (Fig. 2B) (Sun et al, 2019). Interestingly, in response to the depolarizing current steps, the BLA neurons from GF mice showed an increased frequency of action potentials compared to those from SPF mice, indicating that neurons from GF mice were more excitable (two-way ANOVA, interaction: $F_{(10, 180)} = 6.285$, $P < 0.0001$; main effect of current level: $F_{(10, 180)} = 383.4$, $P < 0.0001$; main effect of GF status: $F_{(1, 18)} = 12.48$, $P = 0.0024$) (Fig. 2B,C). Intriguingly, the intrinsic neuronal properties (resting membrane potential (RMP), membrane resistance (Rm), and membrane capacitance (Cm)) of the BLA principal neurons of GF mice were not significantly different from those of SPF mice (Appendix Table S1). While voltage-gated $Na^+$ and $K^+$ channels are important determinants of neuronal

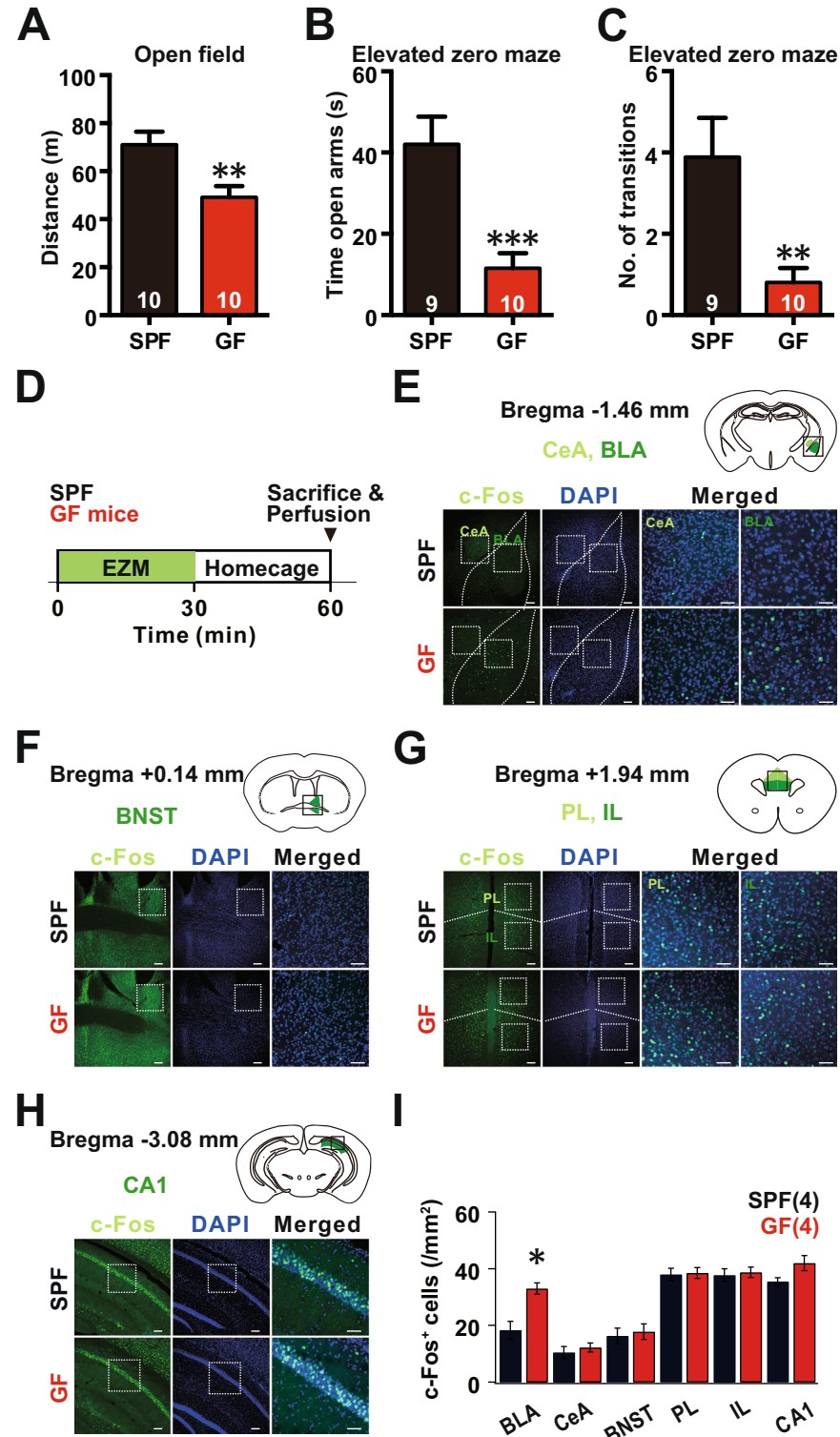

excitability, the current densities of $I_{Na}$ and $I_k$ recorded from BLA principal neurons of GF mice were not significantly different. ($I_{Na}$ two-way ANOVA, Interaction: $F_{(14, 308)} = 0.1275$, $P > 0.9999$; the main effect of voltage steps: $F_{(14, 308)} = 83.89$, $P < 0.0001$; main effect of GF status: $F_{(1, 22)} = 0.0211$, $P = 0.8858$; $I_K$ two-way ANOVA, Interaction: $F_{(14, 308)} = 0.6963$, $P = 0.456$; main effect of voltage

steps: $F_{(14, 308)} = 220.5$, $P < 0.0001$; main effect of GF status: $F_{(1, 22)} = 1.419$, $P = 0.2463$) (Appendix Fig. S2A,B), and no group difference in the amplitude and frequency of spontaneous inhibitory postsynaptic currents (sIPSCs) was observed(frequency: SPF = $12.15 \pm 1.72$ Hz;   GF = $10.24 \pm 1.57$ Hz;   Student's $t$ test,

**Figure 1. Representation of anxiety-related behaviors in germ-free (GF) mice.**

(A) Total distance traveled in the open field test in GF and SPF mice; $P = 0.0067$. (B, C) Total time spent in the open arms of the elevated zero maze (B), and the number of transitions between the two closed arms (C) in GF and SPF mice, SPF $N = 9$, GF $N = 10$; $P = 0.0008$, $P = 0.0061$, respectively. (D) A behavioral schedule. To induce anxiety-related neuronal activation, mice were exposed to an elevated zero maze for 30 min and left undisturbed in their home cages for 30 min before being sacrificed for immunohistochemistry. (E) Activity mapping of the amygdala using c-Fos staining in SPF control and GF mice. The basolateral amygdala (BLA) and the central amygdala (CeA) were defined in a fluorescent image of c-Fos immunoreactivity (green) and nuclear staining with DAPI (blue), based on corresponding brain areas in the Allen Mouse Brain Atlas, as shown above. Areas with the highest difference in c-Fos expression are shown in BLA and CeA. A magnified image of c-Fos immunoreactivity corresponding to the inset in the fluorescence image is shown in "Merged". (F–H) Representative images of c-Fos-positive cells in different brain regions. Schematics were drawn based on the Allen Mouse Brain Atlas. (I) Quantification of the number of c-Fos-positive (+) cells in the SPF control and GF groups ($N = 4$); $P = 0.0001$. PL prelimbic cortex, IL infralimbic cortex, BNST bed nucleus of the stria terminalis, CeA central amygdala, BLA basolateral amygdala. Scale bars: 100 μm in (left) and 50 μm in (right). All Ns are biological replicates. *$P < 0.05$, **$P < 0.01$, ***$P < 0.001$. Data are presented as mean ± SEM. Student's $t$ test and Mann–Whitney test. All mice were aged 10–12 weeks old at the time of testing. Source data are available online for this figure.

$P = 0.4226$; and amplitude: SPF = 41.06 ± 3.68 pA; GF = 47.51 ± 3.31 pA; Student's $t$ test, $P = 0.5592$) (Fig. 2D,E).

Since we observed increased firing in BLA principal neurons of GF mice, we hypothesized that the network activity of these neurons, and thus the action potential-driven spontaneous excitatory postsynaptic currents (sEPSCs), might be increased. We found that the frequency (SPF: 4.29 ± 0.4 Hz; GF: 7.93 ± 1.08 Hz; Student's $t$ test, $P = 0.0077$), but not the amplitude (SPF: 23.41 ± 1.68 pA; GF: 24.64 ± 1.25 pA; Student's $t$ test, $P = 0.5592$) of sEPSCs was increased in GF mice (Fig. 2F,G) The increased sEPSC frequency could be due to an increase in presynaptic release probability resulting from the increased intrinsic excitability of their BLA neurons. To further confirm this, we performed miniature(m)EPSC/mIPSC recording experiments and found that neither the frequency (SPF: 5.81 ± 0.32 Hz; GF: 5.65 ± 0.40 Hz; Mann–Whitney test $P = 0.83$) nor the amplitude (SPF: 24.54 ± 0.22 pA; GF: 21.34 ± 0.33 pA; Mann–Whitney test $P = 0.80$) of mEPSCs was altered in GF mice (Appendix Fig. S2C,D), suggesting that blockade of action potentials with TTX blocks the intrinsic excitability of BLA neurons and that the changes observed in sEPSCs reflect network activities due to increased intrinsic excitability of BLA neurons in the GF, not via the number of presynaptic terminals or presynaptic functional changes such as release probability or quantal content. Similarly, no differences in either amplitudes or frequencies of mIPSCs recorded from BLA neurons were observed between GF and SPF mice (Appendix Fig. S2E,F).

Based on our findings, we concluded that the glutamatergic receptor function was unlikely to be altered between the GF and SPF mice, as there was no difference in the sEPSC amplitude. In addition, the intrinsic property of CeA neurons, which are adjacent to the BLA in GF mice, was not altered (Appendix Fig. S2G,H). Since the current densities of $I_{Na}$ and $I_k$ recorded from BLA principal neurons of GF mice were not significantly different, we investigated other ion channels that are known to be associated with anxiety.

## Decreased activity of SK channels in BLA principal neurons in GF mice

Several mechanisms could account for the increased neuronal excitability in GF mice, including modulation of the sodium-potassium pump, resting K$^+$ conductance, the H current (Ih), or voltage-gated K$^+$ current activated near the subthreshold. However, since increased excitability was observed without significant changes in

RMP and Rm in the GF group, we believe it is less likely that the effect was mediated by inhibition of the sodium-potassium pump or resting K$^+$ conductance. On the other hand, Ih is generated by hyperpolarization-activated cyclic nucleotide-gated (HCN) channels and regulates neuronal excitability by maintaining RMP and AHP. Since HCN channels are expressed in BLA principal neurons, where they regulate neuronal excitability and anxiety (Park et al, 2007), we investigated whether they also contribute to the increased firing in these neurons in GF mice. To measure HCN currents (Ih), we injected currents ranging from −140 mV to −70 mV in the voltage clamp configuration. As can be seen in Fig. 3A,B, Ih in each group was not significantly different (two-way ANOVA, Interaction: $F_{(7, 98)} = 0.6963$, $P < 0.6750$; Main effect of voltage steps: $F_{(7, 98)} = 117.8$, $P < 0.0001$; Main effect of GF status: $F_{(1, 14)} = 1.052$, $P = 0.3225$). These findings suggest that HCN channels do not contribute to the increased firing in BLA principal neurons.

Based on the literature review, we noted a causal relationship between SK channels in BLA principal neurons and anxiety-related behaviors (Morel et al, 2019). Indeed, while chronic stress resulted in hyperexcitability of BLA principal neurons and increased anxiety-related behaviors, we observed a decrease in SK channel-dependent medium afterhyperpolarization (mAHP), consistent with increased excitability (Rau et al, 2015). Thus, these data are consistent with an altered SK channel activity in BLA principal neurons resulting from increased firing activity of these neurons in GF mice.

To test this, we measured the amplitude of AHPs from the spike traces elicited by 50–500 pA current injections (lasting for 1 s), with the membrane potential held at −65 mV. mAHPs were quantified as the peak negative potential after the end of the spike train elicited by the current steps (Fig. 3C), whereas BK channel-dependent fast AHP (fAHP) amplitude was measured as the peak negative potential after the first action potential at the current step that elicited the fewest spikes (Appendix Fig. S3A). We measured mAHP amplitude using traces with similar spike counts in response to the current injection, because mAHP levels are known to be positively related to the number of spikes evoked but not to the magnitude of the current injected (Abel et al, 2004). The amplitude of mAHP evoked by 20 to 25 spikes was included in the analysis. Strikingly, we recorded a significantly reduced mAHP in GF neurons compared with SPF neurons (SPF: −2.98 ± 0.33 mV; GF: −1.56 ± 0.25 mV; Student's $t$ test, $P = 0.0017$) (Fig. 3D). However, we did not observe any group difference in fAHP (Appendix Fig. S3B). To investigate the relationship between SK channels and increased neuronal excitability in GF mice, we tested whether bath application of 1-EBIO, an SK channel agonist, could reverse or prevent this increased excitability.

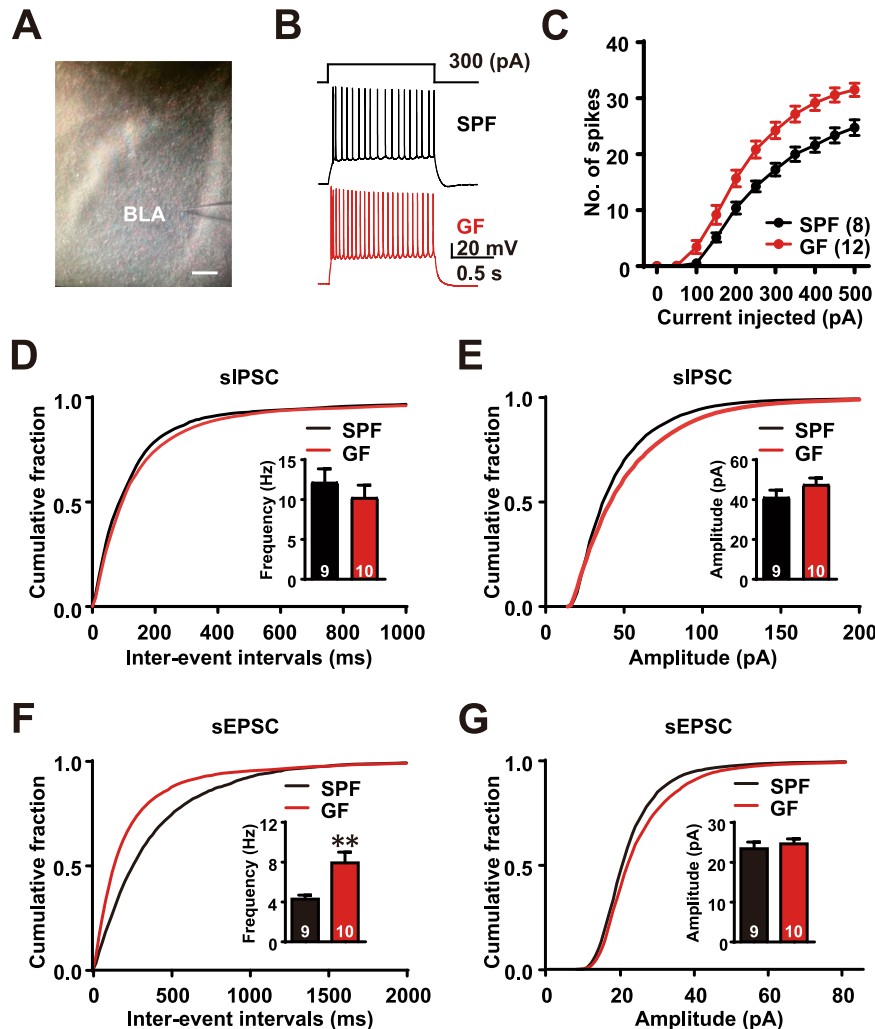

**Figure 2. Increased excitability of principal neurons in the BLA of GF mice.**

(A) A micrograph showing the patch pipette targeting BLA principal neurons. (Scare bar: 0.1 mm. (B) Representative traces showing spike firing in response to 300 pA current injection in BLA neurons from GF and SPF mice. (C) The number of spikes induced by current steps of 50–500 pA in GF and SPF neurons. (D) Cumulative probability distribution of the inter-event interval of sIPSCs and the quantification of sIPSC frequency recorded from BLA neurons of SPF and GF mice. (E) Cumulative probability distribution of the sIPSC amplitude and the quantification of sIPSC amplitude. (F) Cumulative probability distribution of sEPSC inter-event interval and the quantification of sEPSC frequency; $P = 0.0077$. (G) Cumulative probability distribution of sEPSC amplitude and the quantification of the mean sEPSC amplitude. SPF: $n = 9$ cells, $N = 4$ mice; GF: $n = 10$ cells, $N = 5$ mice. All Ns are biological replicates. Data are expressed as mean ± SEM. Two-way ANOVA and Student's t test. **$P < 0.01$. All mice were aged 10–14 weeks old at the time of testing. Source data are available online for this figure.

Interestingly, 1-EBIO (400 μM) treatment reversed the increased firing in response to the current injection in GF mice. (Two-way ANOVA, Interaction: $F_{(10, 180)} = 6.779$, $P < 0.0001$; Main effect of current steps: $F_{(10, 180)} = 369.3$, $P < 0.0001$; Main effect of GF status: $F_{(1, 18)} = 16.25$, $P = 0.0008$) (Fig. 3E,F). To further demonstrate the effect of the SK channel in modulating neuronal firing, we applied apamin to BLA principal neurons in SPF mice and analyzed whether the increased firing in GF mice could be mimicked by blocking the SK channel in SPF mice. While apamin treatment led to an increase in spike frequency in SPF mice (two-way ANOVA, Interaction: $F_{(10, 160)} = 4.036$, $P < 0.0001$; Main effect of current steps: $F_{(10, 160)} = 401.7$, $P < 0.0001$; Main effect of GF status: $F_{(1, 16)} = 10.33$, $P = 0.0054$) (Appendix Fig. S3C,D), no further change in spike frequency was observed in GF mice. (Two-way ANOVA, interaction: $F_{(10, 140)} = 0.8881$, $P < 0.546$; main effect of current steps:

$F_{(10,140)} = 432.2$, $P < 0.0001$; main effect of GF status: $F_{(1, 14)} = 19.93$, $P = 0.3154$) (Appendix Fig. S3E,F). These results confirm that SK channel dysfunction plays a key role in the induction of BLA principal neuron hyperexcitability in microbiota-depleted GF mice. Furthermore, the increased firing in GF mice was reproduced by inhibiting SK channels in SPF mice (Appendix Fig. S3C,D).

## Restoration of the behavioral and physiological defects of the GF mice by conventionalization

To test whether the identified neurophysiological and behavioral changes correlated with the GF status, we inoculated microbiota into GF mice, in a process known as conventionalization (Darnaud et al, 2021). Subsequently, we subjected the CONV and GF mice to

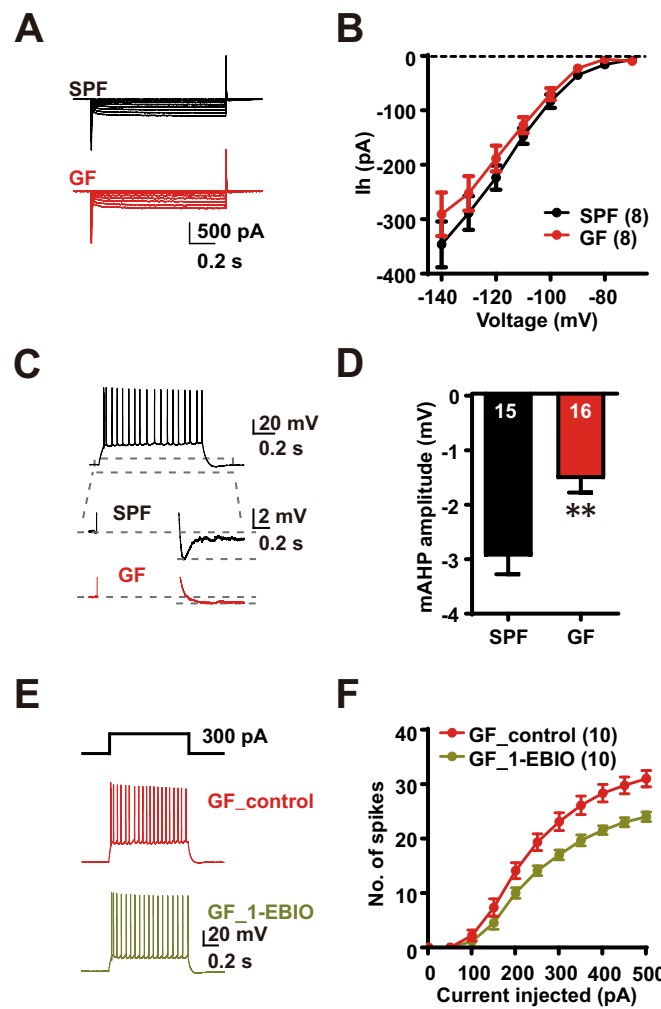

**Figure 3. Decreased activity of SK channels in principal BLA neurons from GF mice.**

(A) Representative traces of Ih current induced by voltage steps ranging from −70 mV to −140 mV in BLA neurons from SPF and GF mice. (B) Quantification of Ih current recorded in SPF and GF BLA neurons. (C) Representative traces of mAHP after spike trains. The dashed line indicates the mAHP amplitude measurements. Scale bar: 5 mV, 0.5 s. (D) Quantification of mAHP amplitude in BLA neurons from SPF and GF mice. SPF $n = 15$ cells $N = 4$ mice, GF $n = 16$ cells $N = 4$ mice; $P = 0.0017$. (E) Representative traces of spike trains induced in BLA neurons from GF mice before and after 1-EBIO application. Scale bar 20 mV, 0.5 s. (F) The number of spikes induced by a 1 s current step ranging from 0 to 500 pA with a 50 pA increment in GF BLA neurons before and after 1-EBIO $n = 10$ cells, $N = 4$ mice. All Ns are biological replicates. Data are expressed as mean ± SEM. Two-way ANOVA and Student's $t$ test. **$P < 0.01$. All mice were aged 10–14 weeks old at the time of testing. Source data are available online for this figure.

behavioral tests, including the open field, and elevated zero mazes, to assess their anxiety-related behaviors. CONV mice showed increased distance traveled in the open field (GF: $61.96 \pm 3.60$ m; CONV: $98.68 \pm 7.167$ m; Student's $t$ test, $P = 0.0002$) as compared to GF mice (Fig. 4A). Next, in the more specific anxiety-related behavioral tests, CONV mice spent more time in the open arms (GF: $11.3 \pm 2.6$ s; CONV: $29.0 \pm 2.8$ s; Mann–Whitney test $P = 0.00169$) (Fig. 4B) and transitioned more frequently (GF: $1.1 \pm 0.62$ s; CONV: $2.4 \pm 0.75$ s; Student's $t$ test, $P = 0.1984$) (Fig. 4C) in the elevated zero maze.

Next, to assess the amygdala activity in CONV mice, we examined c-Fos expression in the BLA area induced by the EZM (Appendix Fig. S4A). Indeed, the density of c-Fos-positive cells in the BLA was significantly reduced in CONV mice (GF: $35.48 \pm 3.979$ n/mm$^2$; CONV: $22.33 \pm 2.858$ n/mm$^2$; Student's $t$ test, $P = 0.0135$) (Fig. 4D). Since CONV mice exhibited normalized intrinsic excitability in BLA neurons, we further tested whether the basal excitatory synaptic transmission was also restored by performing sEPSC recording. Compared to GF mice, sEPSCs from CONV mice displayed decreased frequency (GF: $19.48 \pm 1.42$ pA; CONV: $18.09 \pm 1.23$ mV; Student's $t$ test, $P = 0.4668$) (Fig. 4E) and unchanged amplitude (GF: $5.64 \pm 0.80$ Hz; CONV: $2.07 \pm 0.29$ Hz; Student's $t$ test, $P = 0.0002$) (Appendix Fig. S4B). Furthermore, examination of the intrinsic excitability and AHPs of BLA neurons in CONV mice showed that the action potential firing rate induced by multiple steps (50–500 pA) of current injection was normalized, as the number of action potentials was significantly reduced compared to that in GF mice (two-way ANOVA, interaction: $F_{(10, 200)} = 5.619$, $P < 0.0001$; main effect of current step: $F_{(10, 200)} = 321.1$, $P < 0.0001$; main effect of GF status: $F_{(1, 20)} = 12.39$, $P = 0.0022$) (Fig. 4F,G). According to the multiple $t$ test, the firing rate was decreased in the current steps of 150–500 pA (multiple comparisons with Holm–Sidak correction, $P < 0.05$) (Fig. 4G). We also measured the amplitude of mAHP and fAHP to assess the function of calcium-activated potassium channels. These results confirmed a significant increase in mAHP amplitude, but not in fAHP amplitude, in CONV mice, consistent with restoration of SK channel function (GF: $-1.27 \pm 0.35$ mV; CONV: $-2.83 \pm 0.37$ mV; Student's $t$ test, $P = 0.0058$) (Fig. 4H,I; Appendix Fig. S4C). Taken together, our data demonstrate that the conventionalization of GF mice successfully alleviated neuronal hyperexcitability of the BLA, and thus decreased anxiety-related behavior, in GF mice.

## Microbe-derived indoles reverse the neurophysiological and behavioral changes in GF mice

Gut microbes are known to to produce millions of microbe-derived metabolites including short-chain fatty acids and tryptophan-derived metabolites when exposed to stress. Among these metabolites, indole, a dominant gut microbial metabolite of tryptophan, exhibits several interesting properties including the ability to rapidly be triggered following glucose starvation (Yi et al, 2017). Moreover, indoles cross the BBB and act on the hypothalamus by modulating signals regulating energy homeostasis in the brain (Romani-Perez et al, 2021). Importantly, previous studies have shown that GF mice have lower serum levels of indole derivatives (Wikoff et al, 2009). Most interestingly, reduced cerebrospinal fluid levels of indole derivatives are known to be associated with impaired fear extinction learning in GF mice (Chu et al, 2019). Given that indole supplementation can restore adult neurogenesis in the hippocampus of adult GF mice (Wei et al, 2021), we sought to determine whether orally administered indole (200 μM; 6 weeks) could modulate anxiety-related behaviors in GF mice (Appendix Fig. S5A).

To assess anxiety-related behavior, we performed the open field and elevated zero maze behavioral tests on SPF, indole-treated GF and vehicle-treated GF mice. As can be seen, the GF group showed reduced motor activity, as evidenced by the total distance traveled (SPF: $81.26 \pm 7.06$ m; GF: $58.54 \pm 4.36$; indole-treated GF:

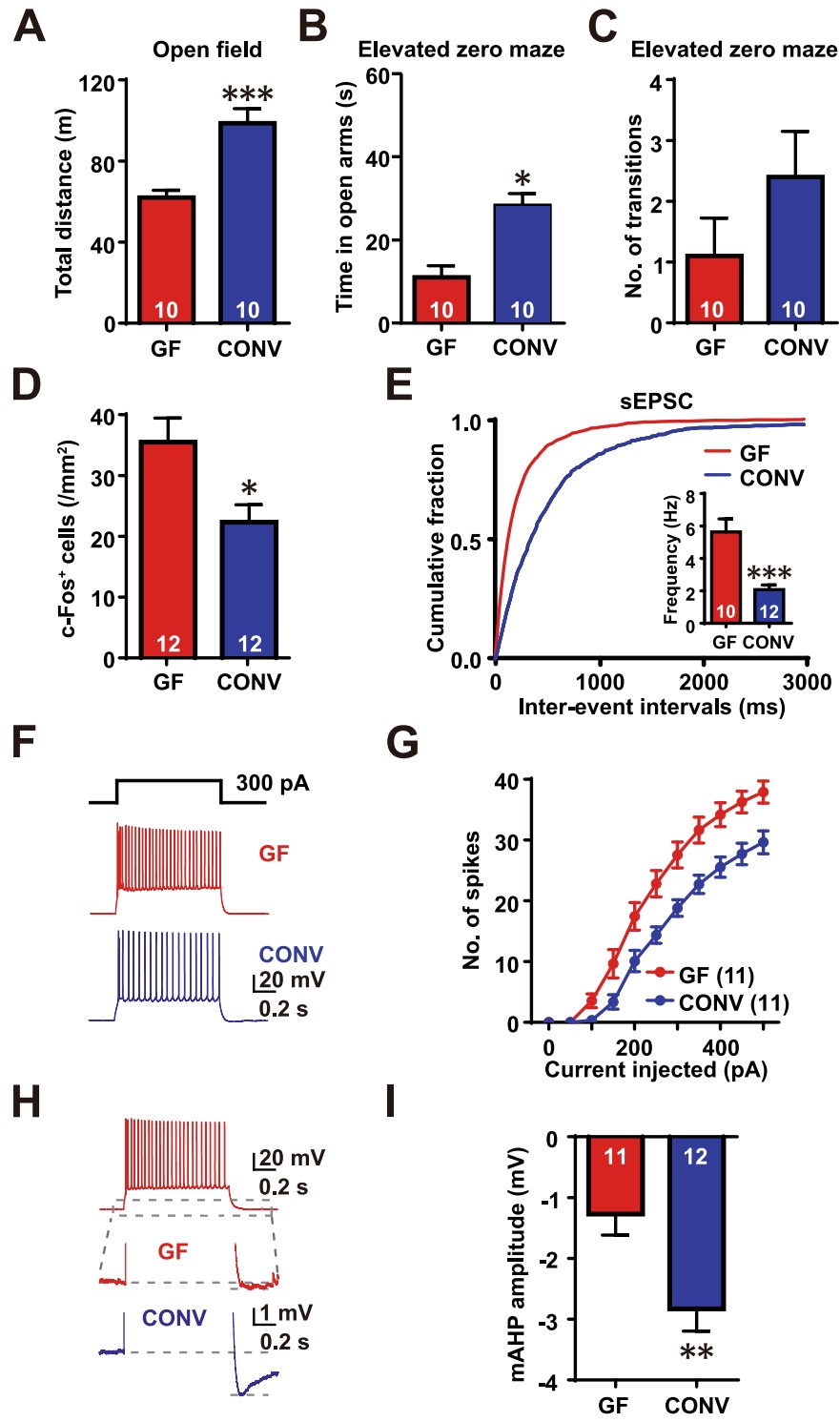

61.68 ± 6.8 m; Student's *t* test, *P* = 0.0163 (SPF vs GF), *P* = 0.067 (SPF vs GF+I), Fig. 5A), compared to the SPF control. To further explore anxiety-like behavior inside the open field, relevant parameters such as the number of central zone entries, the distance traveled in the central zone, and the ratio of central to peripheral zone distances were measured. We also monitored their relative positions using a heat map to represent the time spent in different

areas (Appendix Fig. S5B–D). The indole-treated group showed a significantly higher number of entries into the central zone entries, compared to the control GF mice (SPF control: 83.5 ± 7.41; GF: 48.5 ± 4.4; indole-treated GF: 69.25 ± 7.23; Student's *t* test, *P* = 0.028) (Fig. 5B). Furthermore, indole-exposed GF mice showed a greater total distance in the central zone (SPF: 15.4 ± 1.46 m; GF: 5.171 ± 0.3 m; indole-treated GF: 7.431 ± 0.63 m; Student's *t* test,

◄

**Figure 4. Conventionalization of GF mice reduced hyperexcitability in the BLA and restored their anxiety-related behaviors.**

(A) Total distance traveled in the open field test. (B, C) Total time spent in the open arms (B) and the number of transitions between the two closed arms (C) of the elevated zero maze. (D) Quantification of the density of c-Fos-positive neurons in the BLA area. Density was calculated as the number of c-Fos-positive cells within the BLA/ the area of the BLA (mm²); $P = 0.0002$, $P = 0.00169$ and, $P = 0.0135$, respectively. (E) Cumulative probability distribution of sEPSCs inter-event interval and the quantification of sEPSC frequency; $P = 0.4668$. (F) Representative traces showing spike firing in response to 300 pA current injection in BLA neurons from GF and CONV mice. (G) The number of spikes induced by current steps of 50–500 pA in GF and SPF BLA neurons. GF $n = 11$ cells $N = 4$ mice, CONV $n = 11$ cells $N = 4$ mice. (H) Representative traces showing spike firing in response to 300 pA current injection in BLA neurons from GF and CONV mice. (I) The number of spikes induced by current steps of 50–500 pA in GF and SPF BLA neurons. GF $n = 11$ cells $N = 4$ mice, CONV $n = 12$ cells $N = 4$ mice. All Ns are biological replicates. $*P < 0.05$, $**P < 0.01$, $***P < 0.001$. Data are presented as mean ± SEM. Two-way ANOVA and Student's $t$ test. All mice were aged 10–12 weeks old at the time of testing. Source data are available online for this figure.

$P = 0.0062$) (Fig. 5C) and an increased ratio of central to peripheral zone distance (SPF: $0.24 \pm 0.017$; GF: $0.1233 \pm 0.007$; indole-treated GF: $0.1676 \pm 0.01$; Student's $t$ test, $P = 0.0032$) (Fig. 5D), suggesting significantly less anxiety-like behavior compared to control GF mice. Assessing the elevated zero maze paradigm, indole-treated mice spent significantly increased time in the open arms compared to the control GF mice (SPF: $86.35 \pm 15.36$ s; GF: $35.77 \pm 10.84$ s; indole-treated GF: $79.23 \pm 10.93$; Student's $t$ test; $P = 0.0172$, Fig. 5E) and showed more number of transitions from closed arms to open arms and back to closed arms (SPF: $16.9 \pm 2.01$; GF: $9.5 \pm 0.85$; indole-treated GF: $16.43 \pm 1.41$; Student's $t$ test; $P = 0.002$, Fig. 5F), suggesting increased exploration of the open arms and less anxiety-like behavior.

To assess the amygdala activity in indole-treated germ-free mice, we next examined c-Fos expression induced by the EZM in the BLA area as well as hippocampal CA1 area (Appendix Fig. S5E,F). Indeed, the density of c-Fos-positive cells in the BLA was significantly reduced in CONV mice (GF: $35.48 \pm 3.979$ n/mm²; CONV: $22.33 \pm 2.858$ n/mm²; Student's $t$ test, $P = 0.0135$) (Fig. 4D). GF mice showed a significant increase in c-Fos positive cells in the elevated zero maze, especially in the BLA, but this change was not observed in GF mice under chronic indole treatment (GF: $33.75 \pm 2.53$ n/mm²; GF + I: $22.5 \pm 1.04$ n/mm²; Mann–Whitney test $P = 0.03$) (Appendix Fig. S5E–G).

We further tested whether indole treatment normalized the intrinsic excitability of BLA neurons in GF mice, and found that the action potential firing rate of BLA neurons in indole-treated GF mice was significantly reduced compared with GF mice (two-way ANOVA, interaction: $F_{(10, 200)} = 5.74$, $P < 0.0001$; main effect of current step: $F_{(10, 200)} = 200.36$, $P < 0.0001$; main effect of GF status: $F_{(1, 20)} = 194.88$, $P = P < 0.0001$) (Fig. 5G,H). Furthermore, our multiple $t$ test analysis showed that the firing rate was decreased with current steps of 150–500 pA (multiple comparisons with Holm–Sidak correction, $P < 0.05$) (Fig. 5H). In contrast, indole-treated SPF mice showed no change in the action potential firing rate of BLA neurons (Appendix Fig. S5H,I). We also measured the amplitude of mAHP and fAHP to assess the function of calcium-activated potassium channels and found a significant increase in mAHP amplitude in indole-treated mice (Fig. 5I). These findings are consistent with a significant restoration of SK channel function (control GF: $-1.88 \pm 0.37$ mV; indole-treated GF: $-2.90 \pm 0.30$ mV; Student's $t$ test, $P = 0.0387$), which was further confirmed by analysis of the apamin-sensitive effect (Fig. 5J). Taken together, our data demonstrate that exposure of GF mice to indole treatment reduced anxiety-related behavior and neuronal hyperexcitability of BLA neurons.

## Discussion

Using a combination of behavioral and electrophysiological studies, we have shown that male GF C57/BL6J mice display increased anxiety-related behavior and fear response, likely caused by the hyperexcitability of excitatory neurons in the BLA. In addition, conventionalization, or dietary supplementation with the microbe-derived metabolite, indole, alleviated anxiety-related behavior in GF mice. In this study, we did not examine the contribution of non-excitatory neurons or non-neuronal cells such as microglia in anxiety-related behaviors, because of the auspicious functional changes in BLA pyramidal neurons with a clear reduction in SK2 channel currents. In addition, we did not monitor levels of 5HT1a receptors, which are also known to regulate anxiety-related behaviors (Ishikawa and Shiga, 2017), and did not perform a detailed monitoring of food intake over 24 h. However, we acknowledge that future studies are needed to determine how changes in microbiota-derived metabolites affect cellular activities in the amygdala and related neural circuits involved in anxiety.

Although anxiety is a normal stress response and an important protective mechanism for coping with malnutrition, persistent anxiety symptoms lead to more chronic situations of anxious behavior, including the development of anxiety disorder—a mental illness that has a lifetime incidence of up to 33.7% (Bandelow and Michaelis, 2015). The high rate of comorbidity reported between anxiety disorders and gastrointestinal disorders suggests that further studies are warranted to better understand the role of the gut-brain axis in the development of anxiety (Banerjee et al, 2017). While anxiety disorders usually occur later in life, an anxious state can be experienced at any stage of life without being considered a disorder/illness. Based on our data, we hypothesize that susceptibility to increased anxiety even at an early age and its development into a full-scale disorder later are closely linked to the levels of certain microbiota including tryptophan-metabolizing bacteria/metabolites such as indoles.

There is a massive metabolic switch at birth, involving a huge nutritional transition period, that triggers several central nervous system structures and circuits (Beluska-Turkan et al, 2019). These include activation of the HPA axis and triggering of alarm signals in the hypothalamus and amygdala, which are known to control food intake and stress-related responses, respectively. Notably, postnatal life is also associated with the transfer of maternal microbes that begin to colonize the offspring. This process is mediated by a complex interplay between the incoming maternal microbes and the new host, promoting the establishment of an ecosystem of microbiota-organ communication pathways to optimize nutrient delivery (Pantazi et al, 2023). Importantly, it

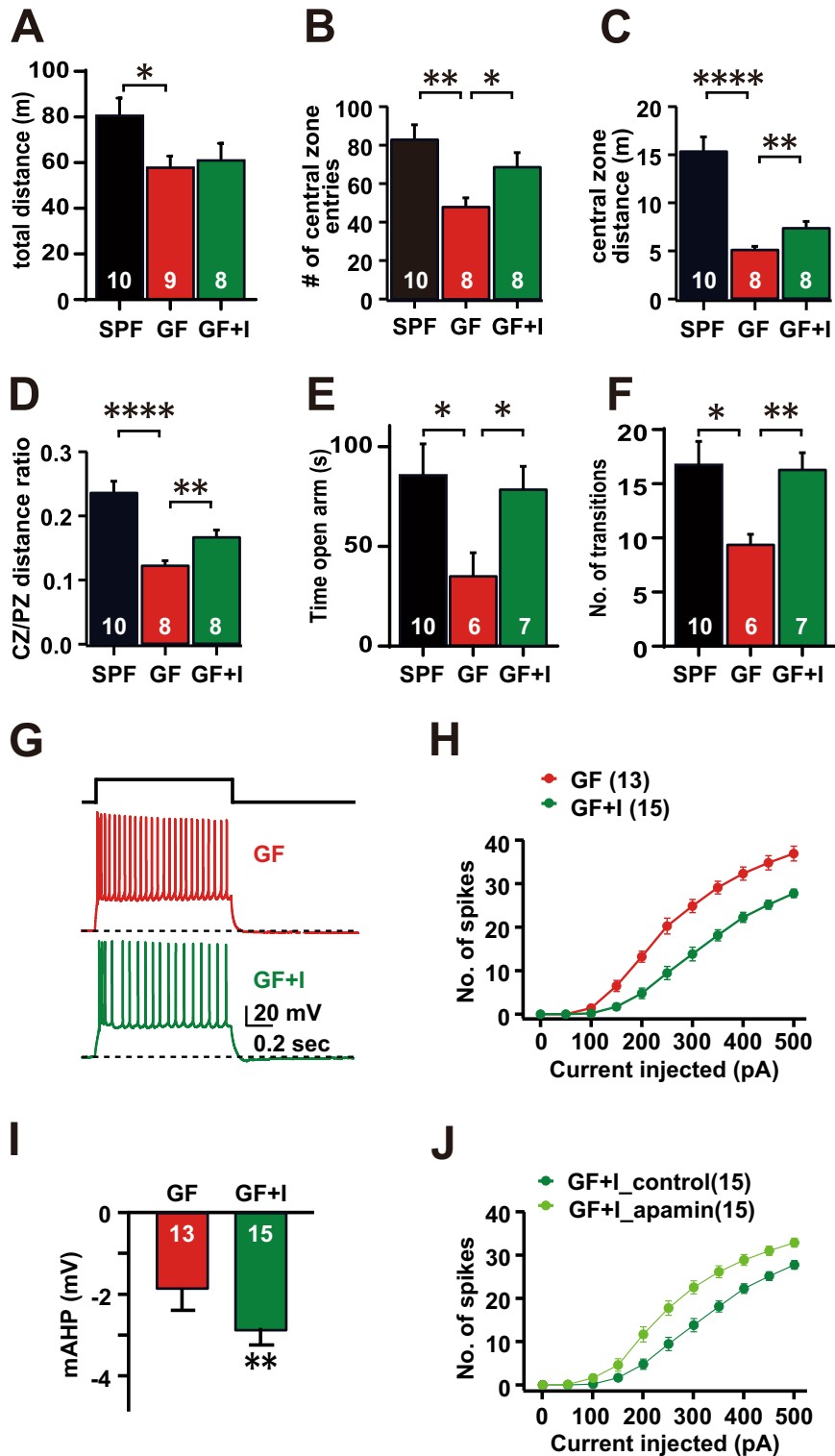

has been reported that breast milk, the initial main source of nutrients for neonatals, contains microbes known to be able to metabolize tryptophan to indoles (van den Elsen et al, 2019). Following the postnatal colonization by maternally-transferred microbes, there is an immediate commitment by the host to manage the massive metabolic switch and modulate its behavioral traits, including nutrition and regulation of host-associated anxiety responses (Sudo et al, 2004). Consistent with this, rodents containing lesions in the BLA or transections of the BLA and the lateral hypothalamus, failed to exhibit improved feeding, further supporting this link (Holland et al, 2002; Petrovich et al, 2002; Holland and Gallagher, 2003). These observations support a role

**Figure 5. Indole treatment reduced anxiety-related behavior and neuronal hyperexcitability of the BLA in germ-free (GF) mice.**

(A) The total distance traveled by SPF, GF, and indole-treated GF (GF + I) mice in the open field test (30 min); $P = 0.067$. (B) The total number of entries into the central zone by SPF, GF, and GF + I mice in the open field test (30 min); $P = 0.028$. (C) The total distance traveled by SPF, GF, and GF + I mice within the central zone of the open field test (30 min); $P = 0.0062$. (D) The ratio of the distance traveled by SPF, GF, and GF + I mice between the central zone and the peripheral zone of the open field throughout the test; $P = 0.0032$. (E) The total time spent by SPF, GF, and GF + I mice in the open arm of elevated zero maze; $P = 0.0172$. (F) The total number of transitions into the open arm in the elevated zero maze by SPF, GF, and GF + I mice; $P = 0.002$. The number of mice per test is indicated on the bars of each graph. (G) Representative traces of spike trains evoked in BLA pyramidal neurons in GF and GF + I mice in response to current injections (300 pA). Scale bar 20 mV, 0.2 s. (H) The number of spikes induced by a 1 s current step ranging from 0 to 500 pA with a 50 pA increment in BLA pyramidal neurons in GF and GF + I mice (GF: $n = 13$, GF + I: $n = 15$, $N = 4$ mice per group). (I) Quantification of mAHP amplitude in BLA neurons in GF and GF + I mice (GF: $n = 13$, GF + I: $n = 15$, $N = 4$ mice per group); $P = 0.0387$. (J) Quantification of the number of spikes induced by a 1 s current step ranging from 0 to 500 pA in BLA neurons from indole-treated GF mice before and after apamin treatment (100 nM). $n = 15$ cells, $N = 4$ mice. All Ns are biological replicates. $^*P < 0.05$, $^{**}P < 0.01$, $^{****}P < 0.0001$. Data are expressed as mean ± SEM. Two-way ANOVA and Student's $t$ test. All mice were aged 12–16 weeks old at the time of testing. Source data are available online for this figure.

for the BLA in various forms of emotional learning related to food (Pavlovian cues) (Johnson et al, 2009). Notably, Pavlovian behavioral instrumental learning does not function when the BLA region is lesioned (Corbit and Balleine, 2005). Based on our data, it is tempting to speculate that susceptibility to increased anxiety even at early age is closely linked to the levels of microbiota/metabolites including tryptophan-metabolizing bacteria/indoles.

Recent research has suggested the role of three potential pathways—the nerve-mediated direct neural connection between the gut and the brain, systemic microbial modulation across the BBB, and endocrine modulation across the BBB by specific microbial metabolites (Kundu et al, 2017; Knox et al, 2022). However, the reported variation in behavioral outcomes in GF mice reported in different studies is likely due to background genetic differences, variation in dietary composition, and degree of infection in the animal house, but remains to be validated (Lu et al, 2018). To address this issue, we generated GF mice using the C57BL/6J strain, which is the most widely used inbred strain and displays the most robust exploratory behavior, best learning and memory performance, and the least anxiety-related behavior (Sultana et al, 2019). Our behavioral tests showed that these C57BL/6J GF mice exhibited increased anxiety-related behaviors (Fig. 1A–C) (Bandelow and Michaelis, 2015). Since the amygdala is known to be the primary center for emotions such as anxiety, fear, and stress response, we monitored the expression of c-Fos, a marker of neural activity, in the BLA pyramidal neurons in GF mice. A single exposure to EZM resulted in increased c-Fos expression relative to its expression levels in SPF mice (Fig. 1E–I). Although we acknowledge that we did not specifically target activated BLA neurons in GF mice using FOS-TRAP transgenic mice, BLA pyramidal neurons in GF mice exhibited increased excitability and firing due to the dysfunction of SK channels, which have been implicated in neuronal adaptation to chronic social defeat stress (Zhang et al, 2019). Genetic background and differential caloric status may impose different levels of stress, as illustrated by a comparison of anxiety in C57/BL6J and NMRI mice (Diaz Heijtz et al, 2011). Moreover, different levels of microbial metabolites present in the diet may further complicate the direct comparison, as may the levels of lipopolysaccharide (LPS), a known mitogen that activates low-grade inflammation (van Eeden et al, 2021). Taken together, these may all be confounding factors that are known to modulate stress hormone levels and brain function.

Live intrinsic microbes and the battery of metabolites produced by them are indispensable components that ensure rapid adaptation of the digestive tract to functional shifts based on nutrient availability (Thriene and Michels, 2023). This intricate metabolic plasticity is an evolutionarily conserved mechanism to minimize unwanted microbial perturbations that may lead to nutritional impairments (Thriene and Michels, 2023). These metabolites act as buffers to meet immediate nutritional needs, and to maintain cellular homeostasis and the integrity of intestinal barriers, including but not limited to the BBB (Braniste et al, 2014). It is well documented that GF mice have reduced levels of indole derivatives in the sera, CSF, and feces (Chu et al, 2019). Indoles can cross the BBB to support adult neurogenesis in the hippocampus and modulate specific neurons in the brain to induce behavioral changes (Wikoff et al, 2009; Chu et al, 2019; Wei et al, 2021). More intriguingly, the defect in fear extinction in GF mice was found to be vagus nerve-independent, suggesting that microbiota-derived metabolites, rather than the direct neuronal connection between the gut and the brain, may be responsible for the behavioral changes in GF mice (Chu et al, 2019). Consistent with this, we found that dietary supplementation with indole restored SK channel function in BLA pyramidal neurons and reduced anxiety-related behavior in GF mice (Fig. 5).

The results presented here suggest that microbial metabolites may be an important future arsenal of factors that will contribute to and play roles in the complex phenomenon we call mammalian anxiety-linked behavior, In addition, the results presented here will have implications for human stress-related diseases, whether for sleep problems or individuals with somatic illnesses who are unable to take standard psychiatric medications. The present study demonstrates and connects how gut microbiota, and their metabolites (indoles) are pivotal in regulating behavioral changes in animals by altering the neuronal excitability of specific neurons. We propose that future studies using indole-fortified foods and or supplementation of indole-producing/tryptophan metabolizing bacteria should be considered as alternative, perhaps more effective, therapeutic strategies for anxiety-related disorders, given that the tryptophan-based diets have shown mixed results (Seltzer et al, 1982; Ohland et al, 2016; Aucoin et al, 2021). This paper consolidates the evolutionary interconnection between microbes, nutrition, and cognition, and the need to always include the microbiota in our attempts to understand eating disorders and anxiety-linked disorders.

# Methods

### Reagents and tools table

| Reagent/resource | Reference or source | Identifier or catalog number |
|---|---|---|
| **Experimental models** | | |
| Germ-free C57BL/6J (*M. musculus*) | In Vivos, Singapore | B6-GF |
| Control C57BL/6J (*M. musculus*) | In Vivos, Singapore | C57BL/6JInv |
| **Recombinant DNA** | | |
| N/A | | |
| **Antibodies** | | |
| Anti-c-Fos | Synaptic systems | 226 017 |
| Anti-NeuN | ThermoFisher | 702022 |
| **Oligonucleotides and other sequence-based reagents** | | |
| N/A | | |
| **Chemicals, enzymes and other reagents** | | |
| Indole | Sigma-Aldrich | I3408 |
| Picrotoxin | Tocris | 1128 |
| Cyanquixaline | Tocris | 1045/1 |
| 1-ethyl-2-benzimidazolinone | Tocris | 1041/10 |
| Apamine | Sigma-Aldrich | 24345-16-2 |
| Tetradotoxin | Abcam | Ab120054 |
| D-AP5 | Sigma-Aldrich | 79055-68-8 |
| NaCl | Sigma-Aldrich | 7647-14-5 |
| KCl | Sigma-Aldrich | 7447-40-7 |
| Glucose | Sigma-Aldrich | 50-99-7 |
| $NaHCO_3$ | Sigma-Aldrich | 144-55-8 |
| $NaH_2PO_4$ | Sigma-Aldrich | 558-80-7 |
| $CaCl_2$ | Sigma-Aldrich | 10035-04-8 |
| $MgCl_2$ | Sigma-Aldrich | 7791-18-6 |
| NMDG | Sigma-Aldrich | 6284-40-8 |
| HCl | Sigma-Aldrich | 7647-01-0 |
| K-gluconate | Sigma-Aldrich | 299-27-4 |
| HEPES | Sigma-Aldrich | 7365-45-9 |
| EGTA | Sigma-Aldrich | 13368-13-3 |
| KOH | Sigma-Aldrich | 1310-58-3 |
| $Na_2ATP$ | Sigma-Aldrich | 4369-07-8 |
| NaGTP | Sigma-Aldrich | 36051-31-7 |
| Sucrose | Sigma-Aldrich | 57-50-1 |
| CsCl | Sigma-Aldrich | 7647-17-8 |
| MgATP | Sigma-Aldrich | 74804-12-9 |
| Biocytin | Sigma-Aldrich | 576-19-2 |
| **Software** | | |
| Any-Maze | Any-Maze, Wood Dale, IL, USA | v5.2 |
| Versamax | AccuScan Instruments, Columbus, Ohio, USA | v4.12 |

| Reagent/resource | Reference or source | Identifier or catalog number |
|---|---|---|
| GraphPad Prism | GraphPad Software, San Diego, CA, USA | v10 |
| ImageJ | NIH, USA | 1.54f |
| Matlab | Matrix Laboratory, The MathWorks, USA | |
| IgorPro | WaveMetrics | v6.37 |
| OriginPro | Microcal | v9.0 |
| pClamp | Molecular Devices, Union City, CA, USA. | V10.4 |
| **Other** | | |
| Open Field Maze | in-house Animal Facility, Singapore | |
| Elevated Zero Maze | in-house Animal Facility, Singapore | |
| Compresstome | Precisionary Instruments, North Carolina, USA | VF-200 |
| Borosilicate glass capillaries | Warner Instruments LLC, Hamden, CT, USA | G150F-4 |
| Axon Multichamp amplifier | Molecular Devices, Union City, CA, USA. | 700B |
| Axon Digidata Digitizer | Molecular Devices, Union City, CA, USA. | 1440 |
| Electrode puller | Narishinge, Japan | PC-10 |
| Compresstome | Precisionary Instruments | VF-200 |
| Fluoromount-G, with DAPI | ThermoFisher | E142380 |

## Animals

Germ-free (GF) and specific pathogen-free (SPF) C57BL/6J male mice aged between 8 and 14 weeks were used. The GF mice were maintained in sterile gnotobiotic plastic isolators at the SingHealth Experimental Medicine Centre and Biological Resource Centre (BRC), A*STAR, Singapore. The sterile status of GF mice was checked weekly by analyzing bacterial cultures of fecal samples from GF cages. All mice were fed autoclaved R36 lactam in their chow (Lactamin). Chow and sterile drinking water were available ad libitum under a 12-h light/dark cycle at 22 °C and 55% humidity. All laboratory animal care and experiments were performed under established protocols and guidelines of the Institutional Animal Care and Use Committee (IACUC), National University of Singapore (IACUC No. 2013/SHS/1276 and 2016/SHS/1210) and BRC (IACUC No. 231784).

## Conventionalization of GF mice

The conventionalized (CONV) mice in this study were the offspring of the colonized GF mice. First, 8-week-old male and female GF mice, breeder mice to produce F1 offspring, were colonized with feces from the SPF mice through a single gavage. They were then left for two weeks before breeding, and the F1 offspring that were born and raised in a normal non-sterile environment were considered CONV mice.

## Indole treatment

Male C57BL/6J mice (8- to 13-week-old GF) were randomly fed regular drinking water or indole-spiked drinking water for 6 weeks. Indole (878.6 mg) was dissolved in Milli-Q water (500 mL, 15 mM) by stirring for 12 h before being filtered twice and diluted in drinking bottles (final concentration of 200 µM) (41). All water was provided ad libitum and changed weekly. All behavior tests were conducted on week 6 of indole treatment. Electrophysiology study was conducted at the end of the 6-week treatment.

## Open field test

The open-field test measures novel environmental exploration and general locomotive behavior (Seibenhener and Wooten, 2015). It also provides an initial assessment of anxiety-related behavior in rodents. The equipment used was a $40 \times 40 \times 30$ cm sized brightly lit arena from Versamax Animal Activity System (AccuScan Instruments, Columbus, Ohio, USA). Activity data were automatically scored by infrared beams of the system, and the distance, mobility time, and center time were analyzed by Versamax software v4.12-1AFE. All open field tests were performed between 9:00 am and 2:00 pm, and a 60-min exploration session was recorded for each mouse. After each session, the number of fecal boils produced by each mouse was scored as an index of anxiety. For the post-indole treatment test session, each animal was placed in the arena and allowed to freely move about for 30 min while being recorded by an overhead camera. The footage was then analyzed by ANY-maze automated tracking system. The major measurements included entries, time, and distance in the central and peripheral zones, indicative of the anxiety levels in rodents. The heat maps were generated from the results using the Heat map report function in the ANY-maze software, set to track the centre point of the mice. A color spectrum was automatically generated based on the time spent by mice at each location —deep blue represents 0 s/minimum time and red represents maximum time (10 s or more) spent at a given location throughout the test.

## Elevated zero mazes

The elevated zero maze apparatus consisted of a blue circular platform with a 6 cm wide path. The platform was 50 cm high from the floor, and the inner diameter was 52 cm. The maze was divided into two open quadrants and two closed quadrants. The closed quadrants were enclosed by 15 cm-high non-transparent walls. A professional video camera was placed right on top of the maze to record the activity inside the maze. TopScan Realtime Option software (Clever Sys, Inc., USA) was used for manual behavioral scoring during the experiments. For each trial, the mouse was first placed in one of the closed quadrants and left for 10 min to allow exploration. The total distance traveled, and the time spent in the open quadrants were measured and analyzed by ANY-maze automated tracking system. Defecation scoring was also conducted after each trial.

## C-Fos immunohistochemistry

The maximum level of c-Fos expression is reported to be induced at 1 to 3 h after acute stimulation (Kovacs, 1998). Therefore, we subjected the test mice to acute stimulation for 1.5 h, before they were sacrificed for c-Fos immunostaining. After EZM, the mice were returned to their home cage and transferred to the perfusion room with a black cover. At 0.5 h post-stimulation, the mice were deeply anesthetized with isoflurane and transcardially perfused with ice-cold phosphate-buffered saline (PBS, pH 7.4 Sigma) for 10 min followed by 4% paraformaldehyde (PFA) in PBS. Brains were retracted and postfixed in 4% PFA PBS solution overnight and thereafter transferred into 30% sucrose (Sigma) solution for cryoprotection. Three days later, the brains were deep-frozen at $-80$ °C until the day of sectioning. 50 µm-sized coronal sections were cut using a Leica cryostat at $-22$ °C, and sections were stored in 4 °C PBS for up to 1 week. To conduct c-Fos immunohistochemistry, the sections were permeabilized with 0.1% Triton X-100 in PBS (PBS-T) for 10 min at room temperature and subsequently blocked with PBS-T buffer containing 2% BSA (Sigma) and 5% donkey serum for 1 h at room temperature. Then, anti-c-Fos polyclonal antibody (1:500, Santa Cruz) and anti-NeuN antibody (1:1000) were diluted in the blocking buffer, and sections were incubated with these primary antibodies for 24 h at 4 °C. The next day, sections were rinsed three times in PBS-T (Sigma). Then, the secondary antibodies, goat anti-mouse antibody conjugated with Alexa 488 (1:1000; ThermoFisher) and donkey anti-rabbit antibody conjugated with Alexa 555 (1:1000; ThermoFisher), were diluted in blocking buffer and applied to the sections for 2 h at room temperature. Then, the sections were rinsed again in PBS-T three times before being mounted on glass slides with FluorSafe mounting medium (Calbiochem). In one batch of experiments, the cell nuclei dye DAPI (1:5000, ThermoFisher) in PBS-T was applied to the sections for 10 min and washed thereafter, before mounting. The c-Fos-stained brain sections were microphotographed with an LSM 710 (Zeiss) confocal microscope that is fitted with a 10x air, 20x air or 40x oil objective, and bilateral c-Fos activation was measured using ImageJ software (National Institute of Health). The expression of e c-Fos protein in BLA neurons was assessed. c-Fos- and NeuN-positive cell counts were bilaterally calculated based on measurements from at least three slices. To quantify the percentage of c-Fos-immunopositive cells, the number of c-Fos-positive cells was counted and divided by the number of NeuN cells, or the area occupied by this structure. The anatomical border of the amygdala subregions was determined manually using ImageJ software, as the subareas were visible in NeuN-stained images obtained from the same slice that was stained for c-fox as well. ImageJ software automatically calculated the number of c-Fos-positive and NeuN-positive nuclei and the area of the subregions.

## Electrophysiology

Slices were obtained from 10- to 14-week-old GF and SPF male mice. In brief, mice were anesthetized with isoflurane inhalation and sacrificed by decapitation following NUS Institutional Animal Care and Use Committee (IACUC) guidelines. The brain was gently removed after opening the skull and submerged into ice-cold artificial CSF (aCSF) consisting of- 124 NaCl, 2.5 KCl, 11 glucose, 25 NaHCO$_3$, 1.25 NaH$_2$PO$_4$, 2 CaCl$_2$, and 1 MgCl$_2$ (in mM). Coronal slices, of 300 µm thickness, were cut on a Compresstome VF-200 (Precisionary Instruments, North Carolina) vibratome in ice-cold aCSF. Slices containing amygdala were transferred to NMDG aCSF consisting of (in mM): 110 NMDG,

110 HCl, 2.5 KCl, 1 NaH$_2$PO$_4$, 25 NaHCO$_3$, 25 glucose, 0.5 CaCl$_2$, and 8 MgSO$_4$ at 33 °C for 12–14 min and then kept in an incubation chamber with normal aCSF at 24 °C to 26 °C for at least 1.5 h before commencing to record. All aCSF solutions used were consistently bubbled with 95% O$_2$ and 5% CO$_2$ to keep the pH approximately 7.4. For whole cell patch clamp recording, slices were transferred to a recording chamber that allowed perfusion of aCSF at 2 ml/min and had temperature controlled at 31 °C –33 °C. Patch electrodes with a 3–5 MΩ resistance were pulled from Borosilicate glass capillaries G150F-4 (Warner Instruments LLC, Hamden, CT) using a PC-10 (Narishige, Japan) electrode puller. All whole-cell patch clamp recordings were performed using an Axon Multiclamp 700B amplifier (Molecular Devices, Union City, CA, USA) and digitized at 10 kHz by an Axon Digidata 1440 Digitizer. Cells were visualized with a 60× water immersion lens on an Olympus BX51 upright microscope equipped with infrared differential inference contrast (IR-DIC) optics. Principal neurons in the basolateral amygdala were identified by their pyramidal-like morphology and verified by their firing pattern, spike width, and AHP amplitude, which are distinct from that of interneurons (Park et al, 2016). To examine intrinsic excitability, the membrane potential of the neurons was kept at −65 mV in the current clamp mood, and then current pulses ranging from 0 to 500 pA in 50 pA increments were injected into the neurons for 1000 ms. Current injections were applied with a 10 s inter-stimulus interval. Picrotoxin (Tocris) (50 μM) and cyanquixaline (CNQX, Tocris) (10 μM) were added to the aCSF to block synaptic responses mediated by GABAA and AMPA receptors, respectively. To test the effects of inhibition and activation of the SK channel on neuronal excitability, the selective SK channel modulators 1-ethyl-2-benzimidazolinone (1-EBIO, 400 μM) and apamin (100 nM) were applied to the slices by direct addition to the perfusing aCSF. To record spontaneous excitatory postsynaptic currents (sEPSCs) and miniature EPSCs (mEPSCs), patch pipette solution containing (in mM) 120 K-gluconate, 10 HEPES, 0.5 EGTA, 9 KCl, 4 NaCl, 10 KOH, 3.48 MgCl$_2$, 4 Na$_2$ATP, 0.4 NaGTP, and 17.5 Sucrose was used. Voltage clamp mode was used, and the membrane potential was held at −70 mV. For isolation of mEPSCs, 1 μM TTX, 50 μM picrotoxin, and 50 μM D-AP5 were added. To record spontaneous inhibitory synaptic currents (sIPSCs) and mIPSCs, patch pipette solution containing (in mM) 140 CsCl, 0.5 CaCl$_2$, 5 EGTA, 10 HEPES, 4 NaCl, 2 MgATP, and 0.4 NaGTP (pH 7.2; 280–290 mOsm) was used and the membrane potential was held at −70 mV. D-AP5 (50 μM) and CNQX (10 μM) were included in the external aCSF. For isolation of mIPSCs, 1 μM TTX, 50 μM picrotoxin, 50 μM D-AP5, and 10 μM CNQX were added. The resting membrane potential (RMP) was measured shortly after membrane rupture. The recording was continued only when the RMP was less than −55 mV, and the access resistance was less than 20 MΩ. Access resistance (typically 5–18 MΩ) was monitored throughout each recording. Cells with more than a 15% change in access resistance were excluded from the analysis. In some recordings, 0.3% biocytin (Sigma) was included in the internal solution to allow better staining and reconstruction of the recorded neurons. The pipette including biocytin was left on the cell for at least 10 min after membrane rupture to ensure complete diffusion of biocytin into the distal dendrites. All the intrinsic excitability data were analyzed by pClamp 10.4 software. The number of spikes induced by each current step was an average of three recordings at a 1 min interval.

## Statistics

The analysis was done blinded, and animals were only referred to by an animal ID (ear tag) during behavior coding or tissue analysis; unmasking was done during statistical testing. Data were analyzed by MATLAB (matrix laboratory, The MathWorks, USA) and GraphPad Prism (GraphPad Software, San Diego, California, USA). Figures were plotted using GraphPad Prism and presented as mean ± SEM (Appendix Table S2). For a single comparison between two groups, two-tailed unpaired $t$ tests were used. Kolmogorov and Smirnov (K-S) tests were used to confirm that the data followed a normal Gaussian distribution, and Bartlett's test was used to test for equality of standard errors. In some experiments, the data did not meet the assumptions of equal variances, either due to a large difference in the number of samples between each group or due to the underlying data itself. In these cases, Student's $t$ test with Welch's correction for unequal variances was used. Multilevel two-factorial comparisons were examined using a mixed factorial design of two-way ANOVA (between-subject factor—GF status, within-subject factor— dependent measures). For the experiments assessing acute drug treatment effects, pre- and post-treatment data were analyzed using two-way ANOVA followed by post hoc analysis (Newman–Keuls test) when appropriate. In some experiments, multiple comparisons of $t$ tests were performed to compare group differences on subsets of the parameter. A Holm–Sidak correction of the $P$ value was performed. A $P < 0.05$ was considered significant. Outliers were identified by the Grubbs outlier test and excluded from the analysis.

---

**The paper explained**

**Problem**

The amygdala, a key brain region that regulates fear and anxiety, has been increasingly linked to the gut-brain axis. However, the precise causal relationship between gut microbes and anxiety-related behaviors remains unclear. In this study, we aimed to assess this relationship by examining the impact of gut microbiota on anxiety-related responses mediated by the amygdala.

**Results**

Our results show that germ-free (GF) mice, which are devoid of gut microbes, exhibit enhanced anxiety-related behaviors. Furthermore, we observed increased action potential firing activities in basolateral amygdala (BLA) pyramidal neurons of GF mice. Interestingly, these neuronal changes were reversed by either conventionalization or dietary indole supplementation.

**Impact**

This study demonstrates how microbiota deficiency induces behavioral changes in animals by altering the excitability of specific neurons. Importantly, these changes can be restored by conventionalization or dietary supplementation with indole, a gut microbe-derived metabolite. This suggests a potential alternative treatment option to reduce anxiety levels in individuals with anxiety-related disorders.

## Data availability

The original contributions presented in the study are included in the article/supplementary material, further inquiries can be directed to the corresponding author/s.

The source data of this paper are collected in the following database record: biostudies:S-SCDT-10_1038-S44321-024-00179-y.

## Peer review information

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

## Acknowledgements

We thank Ms. Norhashimah Sulaiman, Dr. Zhang Wei, Ms Sofia George, and the BRC GF team for their assistance in maintaining the GF animals. We thank Ms. Sruthi Jagannathan for her constructive comments and suggestions. We thank the SingHealth Advanced BioImaging (ABI) and the ABI staff for the training, consultation, advice, and technical support. ABI is partially funded by the National Research Foundation, Singapore under its Shared Infrastructure Support grant for SingaScope (NRF2017_SISFP10). This work was supported by the Singapore Ministry of Education Academic Research Fund (MOE-T2EP30121-0032), National Medical Research Council Open-Fund Individual Research Grant (NMRC-OFIRG21jun-0037), National Research Foundation Competitive Research Programme (NRF-CRP17-2017-04), Duke-NUS Signature Research Program Block Grant (to HSJ), and Young Individual Research Grant (NMRC-OFYIRG22jul-0028, to WY). The funding support from Sunway University, Malaysia; National Neuroscience Institute, Singapore; Dept of Odontology, Karolinska Institute; and Canadian Institute for Advanced Research, Canada (to SP) are also acknowledged.

## Author contributions

**Weonjin Yu**: Conceptualization; Data curation; Formal analysis; Funding acquisition; Investigation; Writing—original draft; Writing—review and editing. **Yixin Xiao**: Conceptualization; Data curation; Formal analysis; Investigation; Writing—original draft. **Anusha Jayaraman**: Conceptualization; Data curation; Formal analysis; Investigation; Writing—original draft; Writing—review and editing. **Yi-Chun Yen**: Formal analysis; Investigation. **Hae Ung Lee**: Formal analysis; Investigation. **Sven Pettersson**: Conceptualization; Supervision; Funding acquisition; Writing—original draft; Writing—review and editing. **H Shawn Je**: Conceptualization; Data curation; Formal analysis; Supervision; Funding acquisition; Investigation; Writing—original draft; Writing—review and editing.

Source data underlying figure panels in this paper may have individual authorship assigned. Where available, figure panel/source data authorship is listed in the following database record: biostudies:S-SCDT-10_1038-S44321-024-00179-y.

## Funding

## Disclosure and competing interests statement

Sven Pettersson is an editorial advisory board member of *EMBO Molecular Medicine*. The remaining authors declare no competing interests.

