## [Peer Review File · EMBO Molecular Medicine]

Microbial metabolites tune amygdala neuronal hyperexcitability and anxiety-linked behaviors

Weonjin Yu, Yixin Xiao, Anusha Jayaraman, Yi-Chun Yen, Hae Lee, Sven Pettersson, and H. Shawn Je

Corresponding author(s): Sven Pettersson (sven.pettersson@ki.se) , H.Shawn Je (shawn.je@duke-nus.edu.sg)

Review Timeline:

Submission Date:	14th May 24
Editorial Decision:	7th Jun 24
Revision Received:	22nd Oct 24
Editorial Decision:	11th Nov 24
Revision Received:	18th Nov 24
Accepted:	18th Nov 24

Editor: Jingyi Hou

Transaction Report:

7th Jun 2024

Dear Sven,

Thank you again for submitting your work to EMBO Molecular Medicine. We have now heard back from the three referees who evaluated your manuscript. As you will see from the reports below, the referees find the topic of your study of interest. However, they raise substantial concerns on your work, which should be convincingly addressed in a major revision of the present manuscript.

The referees' recommendations are relatively straightforward, so there is no need to reiterate the points listed below. All the issues raised by the reviewers need to be carefully addressed. During our pre-decision cross-commenting (in which the referees are given a chance to make additional comments, including on each other's reports), Referees #2 mentioned that their comments #1,4,6,9,12 need to be addressed in a major revision, and the other questions raised by the same referee should be taken seriously and, at the very least, discussed in the DISCUSSION section. Referee #1 concurred with Referee #2 and indicated that the concern of both Referees #1 and #3's regarding sex comparison needs to be addressed.

We would welcome the submission of a revised version within six months for further consideration. Please feel free to contact me in case you would like to discuss in further detail any of the issues raised by the referees.

As you may already know, our editorial policy allows in principle a single round of major revision, and it is therefore essential to provide responses to the reviewers' comments that are as complete as possible.

EMBO Molecular Medicine has a "scooping protection" policy, whereby similar findings that are published by others during review or revision are not a criterion for rejection. Should you decide to submit a revised version, I do ask that you get in touch after six months if you have not completed it, to update us on the status.

I look forward to seeing a revised form of your manuscript as soon as possible.

Sincerely,
Jingyi

Jingyi Hou
Editor
EMBO Molecular Medicine

We require:

- 1) A .docx formatted version of the manuscript text (including legends for main figures, EV figures and tables). Please make sure that the changes are highlighted to be clearly visible.
- 2) Individual production quality figure files as .eps, .tif, .jpg (one file per figure). For guidance, download the 'Figure Guide PDF': (<https://www.embopress.org/page/journal/17574684/authorguide#figureformat>).
- 3) A .docx formatted letter INCLUDING the reviewers' reports and your detailed point-by-point responses to their comments. As part of the EMBO Press transparent editorial process, the point-by-point response is part of the Review Process File (RPF),

which will be published alongside your paper.

- 4) A complete author checklist, which you can download from our author guidelines (<https://www.embopress.org/page/journal/17574684/authorguide#submissionofrevisions>). Please insert information in the checklist that is also reflected in the manuscript. The completed author checklist will also be part of the RPF.
 - 5) Please note that all corresponding authors are required to supply an ORCID ID for their name upon submission of a revised manuscript.
 - 6) It is mandatory to include a 'Data Availability' section after the Materials and Methods. Before submitting your revision, primary datasets produced in this study need to be deposited in an appropriate public database, and the accession numbers and database listed under 'Data Availability'. Please remember to provide a reviewer password if the datasets are not yet public (see <https://www.embopress.org/page/journal/17574684/authorguide#dataavailability>).
- In case you have no data that requires deposition in a public database, please state so in this section. Note that the Data Availability Section is restricted to new primary data that are part of this study.
- 7) For data quantification: please specify the name of the statistical test used to generate error bars and P values, the number (n) of independent experiments (specify technical or biological replicates) underlying each data point and the test used to calculate p-values in each figure legend. The figure legends should contain a basic description of n, P and the test applied. Graphs must include a description of the bars and the error bars (s.d., s.e.m.). See also 'Figure Legend' guidelines: <https://www.embopress.org/page/journal/17574684/authorguide#figureformat>
 - 8) At EMBO Press we ask authors to provide source data for the main manuscript figures. Our source data coordinator will contact you to discuss which figure panels we would need source data for and will also provide you with helpful tips on how to upload and organize the files.
 - 9) Our journal encourages inclusion of *data citations in the reference list* to directly cite datasets that were re-used and obtained from public databases. Data citations in the article text are distinct from normal bibliographical citations and should directly link to the database records from which the data can be accessed. In the main text, data citations are formatted as follows: "Data ref: Smith et al, 2001" or "Data ref: NCBI Sequence Read Archive PRJNA342805, 2017". In the Reference list, data citations must be labeled with "[DATASET]". A data reference must provide the database name, accession number/identifiers and a resolvable link to the landing page from which the data can be accessed at the end of the reference. Further instructions are available at .
 - 10) We replaced Supplementary Information with Expanded View (EV) Figures and Tables that are collapsible/expandable online. A maximum of 5 EV Figures can be typeset. EV Figures should be cited as 'Figure EV1, Figure EV2' etc... in the text and their respective legends should be included in the main text after the legends of regular figures.

.

13) Author contributions: You will be asked to provide CRediT (Contributor Role Taxonomy) terms in the submission system. These replace a narrative author contribution section in the manuscript.

14) A Conflict of Interest statement should be provided in the main text.

Please also suggest a striking image or visual abstract to illustrate your article as a PNG file 550 px wide x 300-800 px high.

**** Reviewer's comments ****

Referee #1 (Remarks for Author):

In this study, Yu and Xiao, et al. reported an interesting finding that C57BL/6J GF mice showed increased anxiety-related behavior compared to C57BL/6J SPF mice. In line with their changes in the behavioral tests, the C57BL/6J GF mice also exhibited increased c-Fos expression and intrinsic excitability in the BLA, which are correlated with reduced SK channel activities. Notably, indole treatment partially rescued the phenotypes of C57BL/6J GF mice. Overall, the findings reported in this manuscript are intriguing and could be informative to other studies in the field, especially since most new transgenic mouse strains are on a C57BL/6J background. However, I have the following concerns:

1. Sex comparison was never analyzed, and it is unclear why only males were used. This is confusing because, in the methods section, the authors wrote, "First, 8-week-old male and female GF mice were colonized with feces from the SPF mice through a single gavage." Did the authors examine female mice or not? Can they provide data on female mice?
2. On page 5, the authors wrote: "We established two distinct control groups: the naive group, in which mice remained undisturbed in their home cages..." but they never showed the data of the naive group.
3. On page 6, the authors wrote, "Interestingly, we observed a significant increase in the number of c-Fos-positive cells in ventral CA1 cells." However, in Figure 11, the CA1 data are comparable between SPF and GF mice.
4. On page 11, Figures S4 and S5 are in the wrong order.
5. What is the total distance traveled by GF vs. GF+I mice in the open field test? What about the time spent in open arms and the number of transitions in the EZM test for GF vs. GF+I mice? Why did the authors choose different parameters when testing GF+I mice? Why didn't they use the same parameters as when they tested SPF vs. GF mice and GF vs. CONV mice in Figures 1 and 4?
6. How is c-Fos expression in the BLA and CA1 of GF+I mice?

Referee #2 (Remarks for Author):

Yu et al.'s study reports elevated BLA neuron activities and anxiety-related behaviors in C57BL/6J GF mice, potentially attributed to the absence of living gut microbiota. However, to conclusively demonstrate that gut microbiota/metabolite dysfunction elevates BLA neuron activities, ultimately leading to anxiety, the provided data is inadequate, and several issues need clarification:

Major Concerns:

1. In Figure 1D, the authors employ the c-fos paradigm in the EZM to identify brain regions involved in anxiety behavior. However, given the statistical data in Figures 1B and 1C, SPF and GF mice react differently in the EZM. This raises questions about whether the observed high c-fos expression level in the BLA of GF mice is due to increased anxiety or simply reflects a longer time spent in the EZM open arm compared to SPF mice.
2. In Figure 1E, it appears that the magnified images of the CeA region for both SPF and GF groups do not correspond to the delimited area in the dotted box.
3. In Figure S1C and S1D, the authors claim no changes in CeA neuron activities. This is surprising given the well-established microcircuit connections between BLA-CeA. The authors should explain why the hyperactivity of BLA neurons did not affect CeA neurons.
4. Figure 1E suggests that not all BLA neurons in GF mice are activated. This raises a crucial question: During the electrophysiological recording of BLA neurons in Figure 2A-C, without any labeling methodology, how can the authors ensure that the recorded neurons were activated? What percentage of recorded neurons were activated BLA neurons? To specifically target activated BLA neurons in GF mice, the use of FOS-TRAP transgenic GF mice would be more appropriate.
5. In Figure 2D-G, sEPSC/sIPSC primarily reflects network activity changes in BLA neurons and may not accurately represent changes in individual BLA neurons. Instead, recording mEPSC/mIPSC would be more insightful.
6. Direct evidence, such as microbiota abundance and diversity analysis, is necessary to confirm the restoration of the gut microbiota community in CONV mice.
7. In the method section (page 17), the authors mention a control group for CONV mice, but no experimental data from this group is presented. Additionally, the SPF group is missing in Figure 4.
8. There is a significant discrepancy in the average time GF mice spent in the open arms of the EZM between Figure 1B (about 10s) and Figure 4B (close to 40s), which is similar to the level of SPF mice in Figure 1B. The authors should explain this substantial variation.
9. There is no direct evidence linking indole to gut microbiota metabolites. To establish this connection, intestinal metabolome analysis of SPF and GF mice is essential.
10. It remains unclear whether 1-EBIO treatment alleviates anxiety behaviors in GF mice.
11. Figure 5D-I lacks a dose-effect curve for indole treatment.
12. Since indole can cross the blood-brain barrier, oral administration of indole may not necessarily affect anxiety-related behaviors through BLA neurons. Direct injection of indole into the BLA would provide stronger evidence.

Minor Concern:

The manuscript requires thorough editing and polishing.

Referee #3 (Comments on Novelty/Model System for Author):

The study is based on the generally accepted fact that murine anxiety like behavior seen in most germ-free murine strains and addresses the underlying mechanisms. Specifically, the authors point out that a causal relationship between the microbiota and anxiety like behavior in rodents remains an association and they wish to establish a causal role of the microbiota. Clinical relevance of the study is also supported by studies showing hyperactivity of the amygdala in patients with anxiety disorders.

The first part of the study demonstrates anxiety-like behavior in C57BL/6J mice; the studies appear to be performed well with clear results. They then show an increase in c-fos positive cells within the basolateral amygdala and demonstrate hyper excitability in the basal lateral amygdala because of a reduction in regulatory K⁺ (SK) activity standard electrophysiological approaches. In the second part of the study, they colonize germ free mice and show that this attenuates, but does not abolish, anxiety like behavior. Microbial colonization also results in reversal of the electrophysiological changes in the BLA. In the final part of the study they administer dietary indole, a microbial metabolite that has been associated with stress and anxiety and is known to cross the blood brain barrier, and show that this improved anxiety like behavior and reverse the hyper excitability in BLA neurons.

This is a novel finding as it strengthens a causal link between microbiota and anxiety like behavior as well as identifies potentially important microbial metabolites involved in this process. The absence of an attempt to identify sex- linked effects in this mechanism is seen as a weakness.

The senior investigators have an established track record related to microbiota- brain interactions.

The model system is standard for this type of work and suits the practical needs of the experiments. However it relates to the early colonization of the gut and may not necessarily relate to anxiety disorders later in life. This should be acknowledged in the revised manuscript.

Referee #3 (Remarks for Author):

This is a well performed study that links the microbiota and its metabolite indole to the regulation of anxiety like behaviour that

characterizes most germ free mouse strains. Anxiety like behaviour is associated with increased excitability in the basolateral amygdala attributed to a reduction and a reduction in calcium-activated potassium (SK) channel currents. Microbial colonization improves but does not abolish anxiety like behaviour but reverses the increased excitability of neurones in the BLA. Dietary indole supplementation also reverses these changes in GF mice.

1. Why was sex-related influence not studied ?

2. Why do the authors feel that colonization only partially reverse anxiety like behaviour, particularly given its impact on the amygdala?

3 Please explain why the electrophysiological experiments performed in the presence of CNQX, and picrotoxin.

4. The authors speculate that these interesting findings might provide insight into anxiety like disorders. However, for practical reasons the study is conducted in the context of early life exposure to microbes and their products, whereas anxiety illness usually occur later in life. this needs to be acknowledged.

5. Do the authors feel that increases in dietary tryptophan may have a role in treating anxiety disorders.

Response to Reviewers

EMBO Molecular Medicine Manuscript # EMM-2024-19971

We thank the reviewers for their helpful comments and suggestions to improve and strengthen our manuscript. To address the concerns raised, we have performed additional experiments and included new data and clarifications in the revised manuscript. The major changes are summarized below, and all changes are described in detail in the point-by-point response following this summary and highlighted in the revised manuscript.

- We have included c-Fos staining data for the naïve groups (Figure S1A-C).
- We have included c-Fos staining data for germ-free (GF) mice, both with and without chronic indole treatment (Figure S5E-G).
- We have added new electrophysiological recordings and analyses of mEPSCs/mIPSCs in the BLA of SPF and GF mice (Figure S2E-H).
- We have added new behavioral assay data and analyses for SPF, GF, and indole-treated GF mice to match with Fig 1 and 4 (Figure 5A, E, F).
- We have clarified the Materials and Methods section.
- We have revised the Discussion section of the manuscript to address the reviewers' comments.

Again, thank you for your valuable feedback.

Reviewer #1

Reviewer #1 found our manuscript "intriguing" and "could be informative for other studies in the field, especially since most new transgenic mouse strains are on a C57BL/6J background". We thank the reviewer for highlighting the importance and impact of our study and have worked to address the comments to further strengthen our manuscript.

Q1. Sex comparison was never analyzed, and it is unclear why only males were used. This is confusing because, in the methods section, the authors wrote, "First, 8-week-old male and female GF mice were colonized with feces from the SPF mice through a single gavage." Did the authors examine female mice or not? Can they provide data on female mice?

The 8-week-old male and female mice mentioned here are the breeder mice (not to be confused with the mice used in the study) - they were colonized and bred with colonized mice to produce F1 offspring. The F1 males were then used throughout the study as this was the original study design. Since the study is focused on behavioral or neurological responses, and the fluctuating hormone levels in the female mice may affect these, interpreting the results difficult and complex, our original plan was to conduct the study in male mice only to gain an initial understanding of the influence without the additional confounding factor of hormone cycling and also to stay within the funded budget. However, we recognize the critical need to compare the sexes, particularly with respect to their responses to stress and anxiety, and the extent to which microbes/metabolites alleviate stress and anxiety between males and females. We believe that this requires separate studies and careful interpretation. Therefore, we have planned a separate study focusing on the influence of microbes/indole on stress and anxiety coping in female mice, with a more nuanced experimental design to account for their hormonal fluctuations, for which we have recently received funding.

Q2. On page 5, the authors wrote: "We established two distinct control groups: the naive group, in which mice remained undisturbed in their home cages..." but they never showed the data of the naive group.

We have agreed with the reviewer to show the data (now in Figure S1 in the revised manuscript). In the naïve group, we did not observe a statistically significant increase in c-Fos positive cells in the BLA, CeA, or CA1 areas within the GF group compared to SPF mice (SPF_BLA: 17.25 ± 1.10 n/mm²; GF_BLA: 22.5 ± 1.32 n/mm²; Mann-Whitney test $P = 0.06$) (Figure S1A, 1B, 1C) (Pg 6, paragraph 1).

Q3. On page 6, the authors wrote, "Interestingly, we observed a significant increase in the number of c-Fos-positive cells in ventral CA1 cells." However, in Figure 11, the CA1 data are comparable between SPF and GF mice.

We thank the reviewer for pointing this out. There is a slight increase in the number of c-Fos positive cells in ventral CA1 cells (Figure 11, SPF: 35.5 ± 1.32 ; GF: 42 ± 2.67 , Mann-Whitney test $P = 0.14$), but not statistically significant. We have corrected this in the revised manuscript (Pg 6. paragraph 3).

Q4. On page 11, Figures S4 and S5 are in the wrong order.

We agree with the reviewer and have revised the manuscript accordingly.

Q5. What is the total distance traveled by GF vs. GF+I mice in the open field test? What about the time spent in open arms and the number of transitions in the EZM test for GF vs. GF+I mice? Why did the authors choose different parameters when testing GF+I mice? Why didn't they use the same parameters as when they tested SPF vs. GF mice and GF vs. CONV mice in Figures 1 and 4?

We agree with the reviewer and have revised Figure 5 in the same format as Figures 1 and 4, adding the total distance graph for the open field and combining the data from Figure 6 into Figure 5 for consistency. We have also revised the figure legend and results section (Pg 12-13) accordingly. When we compared SPF, GF, and GF+I mice together in another independent set to generate data for the revised Figure 5, we added a few more open field parameters relevant to anxiety-like behavior, such as the number of central zone entries, the distance traveled in the central zone, and the ratio of central to peripheral zone distances, which were missing from the previous set (Figures 1 and 4). The total distance traveled in the open field does not always reflect anxiety-like behavior as accurately as where the mice tend to go inside the box. Anxiety-like behavior is usually determined by the extent to which mice explore the central zone, as they tend to stay in the peripheral area close to the walls of the open field chamber when they are anxious, rather than venturing into the center of the chamber.

Q6. How is *c-Fos* expression in the BLA and CA1 of GF+I mice?

Figure 2. Expression of *c-Fos* in neurons of the amygdala and CA1 region in indole-treated mice. (A-B) *c-Fos* staining was used to map neuronal activity in the amygdala and CA1 region of germ-free (GF) mice following the elevated zero maze (EZM) experiment. The most significant differences in *c-Fos* expression were observed in the basolateral amygdala (BLA). A magnified view of the *c-Fos* immunoreactivity, corresponding to the inset in the fluorescence image, is presented in the "Merged" panel. (C) Quantitative analysis of *c-Fos*-positive (+) cells in GF mice and indole-treated GF mice. Scale bars: 100 μ m in (left) and 50 μ m in (right).

We have agreed with the reviewer to perform *c-Fos* staining experiments (now in Figure S5 in the revised manuscript). Compared to SPF mice, GF mice showed a significant increase in *c-Fos* positive cells in the elevated zero maze, especially in the BLA (Figure 1E and G), but this change was not observed with GF mice with chronic indole treatment (GF: 33.75 ± 2.53 n/mm²; GF + I: 22.5 ± 1.04 n/mm²; Mann-Whitney test $P = 0.03$) (Figure S5E-G) (Pg 13, paragraph 2).

Referee #2:

Reviewer #2 raised multiple concerns and we have addressed the comments to further strengthen our manuscript.

Q7. In Figure 1D, the authors employ the c-fos paradigm in the EZM to identify brain regions involved in anxiety behavior. However, given the statistical data in Figures 1B and 1C, SPF and GF mice react differently in the EZM. This raises questions about whether the observed high c-fos expression level in the BLA of GF mice is due to increased anxiety or simply reflects a longer time spent in the EZM open arm compared to SPF mice.

We understand the reviewer's concern and have shown c-Fos staining data in both SPF and GF animals under naive conditions and observed only residual c-Fos staining in both groups in the home cage (please take a look at Figure 1 in Q2 in this rebuttal, now new Figure S1 in the revised manuscript). These results suggest that c-Fos levels in the BLA of GF mice reflect neuronal activity and that this activity is increased by a sensory stimulus or environment that affects anxiety levels, rather than the time spent in the EZM.

Q8. In Figure 1E, it appears that the magnified images of the CeA region for both SPF and GF groups do not correspond to the delimited area in the dotted box.

We agree with the reviewer and have revised the images accordingly (Figure 1E).

Q9. In Figure S1C and S1D, the authors claim no changes in CeA neuron activities. This is surprising given the well-established microcircuit connections between BLA-CeA. The authors should explain why the hyperactivity of BLA neurons did not affect CeA neurons.

We agree with the reviewer that increased excitability of BLA pyramidal neurons would likely facilitate activity in downstream regions such as the prefrontal cortex, central amygdala, hippocampus, and nucleus accumbens. However, the situation may be more complex depending on the internal or external state of the behavior. Previous studies have shown that optogenetic stimulation of BLA terminals in the central amygdala (CeA) produces an acute, reversible anxiolytic effect (Tye et al. 2011), whereas suppression of this projection increases anxiety-related behaviors. This finding seems to contradict the reviewer's prediction. Instead, we believe that the projection from the BLA to the ventral hippocampus (vHPC) is more relevant for mediating anxiety, based on previous optogenetic studies (Felix-Ortiz et al. 2013).

This conclusion is also supported by our c-Fos staining data showing increased c-Fos expression in the ventral hippocampal area after exposure to the elevated plus maze (Figure 3 in the rebuttal, now in Figure 1I in the revised manuscript). However, the situation may be more complicated because BLA inputs exert polysynaptic inhibitory effects on local interneurons in the vHPC. This inhibition may lead to hyperactivity in excitatory principal neurons (Felix-Ortiz et al. 2013). We believe that dissecting these effects is beyond the scope of our current manuscript.

Q10. Figure 1E suggests that not all BLA neurons in GF mice are activated. This raises a crucial question: During the electrophysiological recording of BLA neurons in Figure 2A-

C, without any labeling methodology, how can the authors ensure that the recorded neurons were activated? What percentage of recorded neurons were activated BLA neurons? To specifically target activated BLA neurons in GF mice, the use of FOS-TRAP transgenic GF mice would be more appropriate.

We agree with the reviewer that it would be ideal to use c-Fos TRAP, germ-free mice to label neurons for the electrophysiological recording (Guenther et al. 2013). Unfortunately, after several attempts, we were unable to get them into germ-free. Although we could only pooled average responses from BLA neurons right after the EZM paradigm, we were able to observe the significant functional changes in intrinsic excitability. We expect that the outcome of research using c-Fos TRAP may not be so different from our current data. We have included this point in the Discussion section of our revised manuscript (Pg 15).

Q11. In Figure 2D-G, sEPSC/sIPSC primarily reflects network activity changes in BLA neurons and may not accurately represent changes in individual BLA neurons. Instead, recording mEPSC/mIPSC would be more insightful.

We agreed with the reviewer and performed mEPSC/mIPSC recording experiments (now in Figure S2 of the revised manuscript). We found that neither the frequency (SPF: 5.81 ± 0.32 Hz; GF: 5.65 ± 0.40 Hz; Mann-Whitney test $P = 0.83$) nor the amplitude (SPF: 24.54 ± 0.22 pA; GF: 21.34 ± 0.33 pA; Mann-Whitney test $P = 0.80$) of mEPSCs were altered in GF mice (Figure 4A and B), suggesting that blockade of action potentials with TTX blocks the intrinsic excitability of BLA neurons and that the changes observed in sEPSCs reflect network activities due to increased intrinsic excitability of BLA neurons in the GF, not via the number of presynaptic terminals or presynaptic functional changes such as

release probability or quantal content. Similarly, there is no change in either amplitudes or frequencies of mIPSCs recorded from BLA neurons between GF and SPF (Figure 4C and D). We have now included this data in the revised manuscript (Pg 7 and 8).

Q12. Direct evidence, such as microbiota abundance and diversity analysis, is necessary to confirm the restoration of the gut microbiota community in CONV mice.

Fecal microbiota transplantation (FMT), the transplantation of a donor gut microbiota community into a recipient mouse, is a well-accepted and reliable experimental tool in the

field of microbiome research. In this study, we performed FMT using feces from donor SPF mice into recipient GF mice. Although we did not perform microbiota profiling for this set of mice, our group has performed FMT and confirmed the restoration of the donor microbiota community in recipient mice in previous studies (Kundu et al. 2019). We have also shown that the conventionalization of mice, similar to what we have done in this study, normalizes/restores several parameters such as metabolite profiles, behavior, and gene expression in GF mice similar to that of regular SPF mice (Lahiri et al. 2019).

Q13. In the method section (page 17), the authors mention a control group for CONV mice, but no experimental data from this group is presented. Additionally, the SPF group is missing in Figure 4.

We agree with the reviewer and have revised the manuscript accordingly (page 17). We did not include the SPF group because there was no difference in behavior after fecal transplantation.

Q14. There is a significant discrepancy in the average time GF mice spent in the open arms of the EZM between Figure 1B (about 10s) and Figure 4B (close to 40s), which is similar to the level of SPF mice in Figure 1B. The authors should explain this substantial variation.

We thank the reviewer for pointing out our error. We have corrected the value and replaced it with a new figure and quantification (Figure 4B in the revised manuscript).

Q15. There is no direct evidence linking indole to gut microbiota metabolites. To establish this connection, intestinal metabolome analysis of SPF and GF mice is essential.

We would like to clarify that indole and its derivatives are themselves metabolites of the gut microbiota. Our gut microbes make indoles by metabolizing dietary tryptophan. Our eukaryotic cells cannot synthesize them. We have previously shown by LC-MS that key metabolites of the tryptophan pathway, such as indoleacetic acid, are significantly lower in GF mice compared to SPF mice (Xing et al. 2024).

Q16. It remains unclear whether 1-EBIO treatment alleviates anxiety behaviors in GF mice.

Rau et al. reported using a rodent model of chronic early-life stress that induces a robust and persistent increase in anxiety-like behaviors (Rau et al. 2015). They found a similar mechanism, noting an increase in intrinsic excitability of BLA pyramidal cells due to decreased expression of SK channels. Rau et al. also performed bath application of a positive SK channel modulator, 1-EBIO, which normalized firing rates in *ex vivo* recordings. In addition, *in vivo* infusion of 1-EBIO into the BLA reduced anxiety-like behaviors.

Q17. Figure 5D-I lacks a dose-effect curve for indole treatment.

We have previously optimized the dosage of indoles both *in vitro* and *in vivo* and demonstrated effects on adult neurogenesis (Wei et al. 2021). We have used the same dose in subsequent projects with indoles, including this study. In addition, our IACUC committee emphasizes reducing the number of mice used in experiments unless there is a strong justification for increasing the number.

Q18. Since indole can cross the blood-brain barrier, oral administration of indole may not necessarily affect anxiety-related behaviors through BLA neurons. Direct injection of indole into the BLA would provide stronger evidence.

Since indoles have been shown to cross the BBB (Pappolla et al. 2021), our approach shows that orally ingested indole affects anxiety by acting on the BLA by crossing the BBB. This is further demonstrated by blocking SK2 channels using electrophysiology. Also from a therapeutic point of view, oral consumption of indoles in the form of fortified diets or supplements or probiotics would be more practical. Therefore, it is important to show the effect of systemic indole on the BLA rather than direct injection into the BLA. In addition, we have previously shown that orally ingested indole is able to cross the BBB and activate its ligand, the aryl hydrocarbon receptor, in the brain (hippocampus) and promote target gene expression *in vivo* (Wei et al. 2021). In addition, others have shown correlations between the lack of indole metabolites in GF mice and behavioral outcomes such as impaired fear extinction learning (Chu et al. 2019).

Q19. The manuscript requires thorough editing and polishing.

We have revised the manuscript according to the reviewer's comments by extensive editing and polishing.

Referee #3

Reviewer #3 considered our manuscript as “a novel finding” that “strengthens a causal link between microbiota and anxiety-like behavior as well as identifies potentially important microbial metabolites involved in this process.” We thank the reviewer for highlighting the significance and impact of our study, and have worked to address the comments to further strengthen our manuscript.

Q20. Why was sex-related influence not studied ?

Please see Q1 in the first reviewer.

Q21. Why do the authors feel that colonization only partially reverse anxiety like behaviour, particularly given its impact on the amygdala?

In our new batch of GF animals, we found that indole treatment reversed anxiety behavior much more robustly than in the previous batch (Figure 5A-F in a revised manuscript). In addition, the downstream pathway for behavioral response to amygdala effects may require other microbes/metabolites in addition to indole. Therefore, responses may be variable in indole-treated GF mice.

Q22 Please explain why the electrophysiological experiments performed in the presence of CNQX, and picrotoxin.

Please see Q11 in the second reviewer.

Q23. The authors speculate that these interesting findings might provide insight into anxiety like disorders. However, for practical reasons the study is conducted in the context of early life exposure to microbes and their products, whereas anxiety illness usually occur later in life. this needs to be acknowledged.

We agree with the reviewer that anxiety disorders are usually associated with later life. However, anxiety is present/felt at any stage of life without being considered a disorder/illness. Furthermore, we hypothesize that the susceptibility to increased anxiety even at an early age, and its development into a full-blown disorder, later on, is closely related to the levels of certain microbiota/metabolites such as tryptophan-metabolizing bacteria/indoles. We have added the following sentences to acknowledge this, as suggested by the reviewer. (Pg 14, paragraph 2, Discussion section): "While anxiety disorders usually occur later in life, an anxious state can be experienced at any stage of life without being considered a disorder/illness. Based on our data, we hypothesize that susceptibility to increased anxiety even at an early age and its development into a full-scale disorder later are closely linked to the levels of certain microbiota including tryptophan-metabolizing bacteria/metabolites such as indoles."

Q24. Do the authors feel that increases in dietary tryptophan may have a role in treating anxiety disorders.

High levels of anxiety have been associated with inadequate dietary tryptophan, and conversely, a tryptophan diet has been shown to reduce anxiety (Aucoin et al. 2021). However, these studies are limited and have been associated with side effects (Seltzer et al. 1982). The efficacy of dietary tryptophan in reducing anxiety appears to depend on the effective processing or metabolism of tryptophan in the gut (Ohland et al. 2016). Alternatively, we suggest that fortification with tryptophan-metabolizing/indole-producing bacteria (probiotics) or indole-enriched foods may be a more beneficial option - to circumvent any negative effects of tryptophan diets and for those with dietary restrictions (vegan/vegetarian, nut allergies) that limit intake of dietary tryptophan. We have added the following sentences discussing this point in the Discussion section (Pg 17, concluding section): "We propose that future studies including use of indole-fortified foods and or supplementation of indole-producing/tryptophan metabolizing bacteria should be considered as alternative therapeutic strategies for anxiety-related disorders even though tryptophan-based diets have given mixed results (59-61). This paper consolidates the evolutionary interconnection between microbes, nutrition, and cognition, and the need to always include the microbiota in our attempts to understand eating disorders and anxiety linked disorders."

References:

- Aucoin, M., L. LaChance, U. Naidoo, D. Remy, T. Shekdar, N. Sayar, V. Cardozo, T. Rawana, I. Chan, and K. Cooley. 2021. 'Diet and Anxiety: A Scoping Review', *Nutrients*, 13.
- Chu, C., M. H. Murdock, D. Jing, T. H. Won, H. Chung, A. M. Kressel, T. Tsaava, M. E. Addorisio, G. G. Putzel, L. Zhou, N. J. Bessman, R. Yang, S. Moriyama, C. N. Parkhurst, A. Li, H. C. Meyer, F. Teng, S. S. Chavan, K. J. Tracey, A. Regev, F. C. Schroeder, F. S. Lee, C. Liston, and D. Artis. 2019. 'The microbiota regulate neuronal function and fear extinction learning', *Nature*, 574: 543-48.
- Felix-Ortiz, A. C., A. Beyeler, C. Seo, C. A. Leppla, C. P. Wildes, and K. M. Tye. 2013. 'BLA to vHPC inputs modulate anxiety-related behaviors', *Neuron*, 79: 658-64.
- Guenther, C. J., K. Miyamichi, H. H. Yang, H. C. Heller, and L. Luo. 2013. 'Permanent genetic access to transiently active neurons via TRAP: targeted recombination in active populations', *Neuron*, 78: 773-84.
- Kundu, P., H. U. Lee, I. Garcia-Perez, E. X. Y. Tay, H. Kim, L. E. Faylon, K. A. Martin, R. Purbojati, D. I. Drautz-Moses, S. Ghosh, J. K. Nicholson, S. Schuster, E. Holmes, and S. Pettersson. 2019. 'Neurogenesis and longevity signaling in young germ-free mice transplanted with the gut microbiota of old mice', *Sci Transl Med*, 11.
- Lahiri, S., H. Kim, I. Garcia-Perez, M. M. Reza, K. A. Martin, P. Kundu, L. M. Cox, J. Selkrig, J. M. Poma, H. Zhang, P. Padmanabhan, C. Moret, B. Gulyas, M. J. Blaser, J. Auwerx, E. Holmes, J. Nicholson, W. Wahli, and S. Pettersson. 2019. 'The gut microbiota influences skeletal muscle mass and function in mice', *Sci Transl Med*, 11.
- Ohland, C. L., E. Pankiv, G. Baker, and K. L. Madsen. 2016. 'Western diet-induced anxiolytic effects in mice are associated with alterations in tryptophan metabolism', *Nutr Neurosci*, 19: 337-45.
- Pappolla, M. A., G. Perry, X. Fang, M. Zagorski, K. Sambamurti, and B. Poeggeler. 2021. 'Indoles as essential mediators in the gut-brain axis. Their role in Alzheimer's disease', *Neurobiol Dis*, 156: 105403.
- Rau, A. R., A. M. Chappell, T. R. Butler, O. J. Ariwodola, and J. L. Weiner. 2015. 'Increased Basolateral Amygdala Pyramidal Cell Excitability May Contribute to the Anxiogenic Phenotype Induced by Chronic Early-Life Stress', *J Neurosci*, 35: 9730-40.
- Seltzer, S., D. Dewart, R. L. Pollack, and E. Jackson. 1982. 'The effects of dietary tryptophan on chronic maxillofacial pain and experimental pain tolerance', *J Psychiatr Res*, 17: 181-6.
- Tye, K. M., R. Prakash, S. Y. Kim, L. E. Fenno, L. Grosenick, H. Zarabi, K. R. Thompson, V. Gradinaru, C. Ramakrishnan, and K. Deisseroth. 2011. 'Amygdala circuitry mediating reversible and bidirectional control of anxiety', *Nature*, 471: 358-62.
- Wei, G. Z., K. A. Martin, P. Y. Xing, R. Agrawal, L. Whiley, T. K. Wood, S. Hejndorf, Y. Z. Ng, J. Z. Y. Low, J. Rossant, R. Nechanitzky, E. Holmes, J. K. Nicholson, E. K. Tan, P. M. Matthews, and S. Pettersson. 2021. 'Tryptophan-metabolizing gut microbes regulate adult neurogenesis via the aryl hydrocarbon receptor', *Proc Natl Acad Sci U S A*, 118.
- Xing, P. Y., R. Agrawal, A. Jayaraman, K. A. Martin, G. W. Zhang, E. L. Ngu, L. E. Faylon, S. Kjelleberg, S. A. Rice, Y. Wang, A. T. Bello, E. Holmes, J. K. Nicholson, L. Whiley, and S. Pettersson. 2024. 'Microbial Indoles: Key Regulators of Organ Growth and Metabolic Function', *Microorganisms*, 12.

11th Nov 2024

Dear Sven,

Thank you for submitting your revised manuscript to EMBO Molecular Medicine. We have now received the enclosed report from the three referees who re-assessed your work. As you will see, the referees are now supportive, and I am pleased to inform you that we will be able to accept your manuscript pending the following amendments:

1. Please move Data availability section to the end of 'Methods' section.
2. Please reduce the keyword number to five.
3. Funding information : Please ensure 'National Neuroscience Institute, Singapore and UK Dementia Group' are included in the online submission system.
4. Please remove the "Authors' contribution" section from the manuscript file.
5. The current "synopsis" section should be renamed to "the paper explained".
6. Please rename "conflict of interest" to " Disclosure statement and competing interests" and add "Sven Pettersson is an editorial advisory board member of EMBO Molecular Medicine".
7. The references need to be formatted according to the EMBO Molecular Medicine reference style:
 - Please list up to 10 co-authors of a paper before adding et al. to the reference list.
 - Citations should be listed in alphabetical order.
 - Please remove DOI for published papers.
8. We recently started to require all Materials and Methods to be presented in the main text using our 'Structured Methods' format. According to this format, the Methods section includes a Reagents and Tools Table (listing key reagents, experimental models, software and relevant equipment and including their sources and relevant identifiers) followed by a Methods and Protocols section describing the methods, ideally using a step-by-step protocol format. The aim is to facilitate adoption of the methodologies across labs.

Please download and fill our Reagents and Tools Table template (.docx), which you can find in our author guidelines:

<https://www.embopress.org/page/journal/17574684/authorguide#structuredmethods>

When submitting your revised manuscript, please do not include the Reagents and Tools Table in the Methods section of the manuscript but upload it as a separate file (.docx) choosing the file type "Reagent Table". The Reagents and Tools Table can be downloaded from our author guidelines (<https://www.embopress.org/page/journal/17574684/authorguide#structuredmethods>)

9. Please provide a 'Synopsis' to further enhance discoverability. Synopses are displayed on the journal webpage and are freely accessible to all readers. They include a short stand first (maximum of 300 characters, including space) as well as 2-5 one-sentences bullet points that summarizes the paper. Please write the bullet points to summarize the key NEW findings. They should be designed to be complementary to the abstract - i.e. not repeat the same text. We encourage inclusion of key acronyms and quantitative information (maximum of 30 words / bullet point). Please use the passive voice. Please attach these in a separate file or send them by email, we will incorporate them accordingly.

Please also provide a visual abstract to illustrate your article as a PNG file 550 px wide x 300-800 px high.

10. Appendix: please correct the nomenclature to "Appendix Table S1" and "Appendix Figure S1" etc. throughout the appendix file. Move the legends underneath the corresponding figure, if possible.
11. Figure legends:
 - Please note that the exact p values are not provided in the legends of figures 1a-c, i; 2f; 3d; 4a-b, d-e, i; 5a-f, i.
 - Please note that information related to n is missing in the legend of figure 1i.

I look forward to reading a new revised version of your manuscript as soon as possible.

Kind regards,
Jingyi

Jingyi Hou
Editor
EMBO Molecular Medicine

*** Instructions to submit your revised manuscript ***

- 1) a .docx formatted version of the manuscript text (including Figure legends and tables)
 - 2) Separate figure files*
 - 3) supplemental information as Expanded View and/or Appendix. Please carefully check the authors guidelines for formatting Expanded view and Appendix figures and tables at <https://www.embopress.org/page/journal/17574684/authorguide#expandedview>
 - 4) a letter INCLUDING the reviewer's reports and your detailed responses to their comments (as Word file).
 - 5) The paper explained: EMBO Molecular Medicine articles are accompanied by a summary of the articles to emphasize the major findings in the paper and their medical implications for the non-specialist reader. Please provide a draft summary of your article highlighting
 - the medical issue you are addressing,
 - the results obtained and
 - their clinical impact.This may be edited to ensure that readers understand the significance and context of the research. Please refer to any of our published articles for an example.
 - 6) Author contributions: the contribution of every author must be detailed in a separate section.
 - 7) EMBO Molecular Medicine now requires a complete author checklist (<https://www.embopress.org/page/journal/17574684/authorguide>) to be submitted with all revised manuscripts. Please use the checklist as guideline for the sort of information we need WITHIN the manuscript. The checklist should only be filled with page numbers where the information can be found. This is particularly important for animal reporting, antibody dilutions (missing) and exact values and n that should be indicated instead of a range.
 - 8) Every published paper now includes a 'Synopsis' to further enhance discoverability. Synopses are displayed on the journal webpage and are freely accessible to all readers. They include a short stand first (maximum of 300 characters, including space) as well as 2-5 one sentence bullet points that summarise the paper. Please write the bullet points to summarise the key NEW findings. They should be designed to be complementary to the abstract - i.e. not repeat the same text. We encourage inclusion of key acronyms and quantitative information (maximum of 30 words / bullet point). Please use the passive voice. Please attach these in a separate file or send them by email, we will incorporate them accordingly.
- You are also welcome to suggest a striking image or visual abstract to illustrate your article. If you do please provide a jpeg file 550 px-wide x 300-600px high.
- 9) A Conflict of Interest statement should be provided in the main text

10) Please note that we now mandate that all corresponding authors list an ORCID digital identifier. This takes <90 seconds to complete. We encourage all authors to supply an ORCID identifier, which will be linked to their name for unambiguous name identification.

Currently, our records indicate that the ORCID for your account is 0000-0001-8903-3738.

Link Not Available

11) Include a Reagents and Tools Table as part of the Methods section, which can be downloaded from our author guidelines (<https://www.embopress.org/page/journal/17574684/authorguide#structuredmethods>)

Photos 400-800 DPI

*Additional important information regarding figures and illustrations can be found at

<https://bit.ly/EMBOPressFigurePreparationGuideline>. See also figure legend preparation guidelines:

<https://www.embopress.org/page/journal/17574684/authorguide#figureformat>

***** Reviewer's comments *****

Referee #1 (Remarks for Author):

I believe the majority of my concerns have been addressed in the revised manuscript. It would have been better if they had included data from female mice at the beginning.

Referee #2 (Comments on Novelty/Model System for Author):

The study conducted by Yu et al. and co-authors reported that C57BL/6J GF mice exhibited elevated BLA neuron activities and anxiety-associated behaviors, and these observed phenomena may be attributed to the lack of living gut microbiota. Although several major concerns existed initially, the authors have addressed these concerns adequately, and the revised manuscript is suitable for publication on EMBO Molecular Medicine.

Referee #2 (Remarks for Author):

All concerns have been addressed.

Referee #3 (Remarks for Author):

I have reviewed the manuscript and the authors response to all three reviewers. I am satisfied that the manuscript is now acceptable for publication.

All editorial and formatting issues were resolved by the authors.

18th Nov 2024

Dear Sven,

I am pleased to inform you that your manuscript is accepted for publication and is now being sent to our publisher to be included in the next available issue of EMBO Molecular Medicine. Thank you for your thorough response to the referee's comments. It has been a pleasure working with you to bring the paper to the acceptance stage.

Kind regards,
Jingyi

Jingyi Hou
Editor
EMBO Molecular Medicine
